

# Sensitivity analysis of point and parametric pedotransfer functions for estimating soil water retention

**Sami Touil [a,b,c], Aurore Degre [b], Mohamed Nacer Chabaca [a]**

[a]*Superior National School of Agronomy, El Harrach, Algiers, Algeria*
[b]*Gembloux Agro-Bio Tech, Biosystem Engineering (BIOSE), Soil—Water—Plant Exchanges, University of Liège, Passage des Déportés 2,Gembloux 5030, Belgium*
[c]*Laboratory of Crop Production and Sustainable Valorization of Natural Resources, University of Djilali Bounaama Khemis Miliana, Ain Defla, Algeria.*

*Correspondence to: Sami Touil (touil_sy@hotmail.fr)*

## Abstract

Improving the accuracy of pedotransfer functions (PTFs) requires studying how the uncertainty in prediction can be apportioned to different sources of uncertainty in inputs. In this study, the question is, which variable input is the principal one or the better complimentary predictor of water retention, and at which water potential? Two approaches were adopted to generate PTF—multiple linear regressions (MLR) for point PTF and the multiple non-linear regressions (MNLR) for parametric PTFs. Reliability tests show that the point PTF provides better estimates than the parametric model (RMSE: 0.0414; 0.0444 and 0.0613; 0.0605 at -33 kPa and -1500 kPa, respectively). The local parametric PTFs provided better estimates than Rosetta PTFs at – 33 kPa. However, no significant difference in accuracy was found between the parametric PTFs and Rosetta-H2 at -1500 kPa with RMSE values (0.0605 $cm^3$ $cm^{-3}$ and 0.0636 $cm^3$ $cm^{-3}$). The results of the global sensitivity analyses show that the mathematical formalism of PTF and its input variables react differently in terms of point pressure and texture. The point and parametric PTFs are sensitive primarily to the sand fraction in fine and medium textural classes. A favourable impact of bulk density and clay content is recorded with accuracy in the estimation of PTFs at -33 kPa in the medium class.

**Keywords:** soil water retention, multiple regressions, pedotransfer function, sensitivity

## I.  Introduction

The predictive information on the spatial distribution of soil water and its availability for plants will allow producers to take effective decisions to maximise profitability (e.g. nutrient management and plant cover). The soil-water balance is in the centre of many processes that influence plant growth and the degradation of soil and water resources. However, due to high temporal and spatial variability in the hydraulic characteristics, a large number of samples are generally required to accurately characterise field conditions (Khodaverdiloo et al., 2011).

Hydrologists are often faced with the situation where one or more of the pedotransfer functions (PTFs) inputs are not available. Reports on the evaluation of PTFs outside the area of development are rare, and the results reveal that extrapolation of PTF in different agropedoclimatic context is limited (Touil et al., 2016). The development of local PTF could be useful to meet the agricultural requirements for modelling with reasonable accuracy.



Soil water retention curves can usually be constructed by two approaches: the point and parameter PTFs. In the point PTFs, the soil water retention is estimated at defined pressure points (Pachepsky et al., 1996; Minasny et al., 1999). The parameterisation method estimates the parameters of soil water retention models such as $\Theta_s$, $\Theta_r$, $\alpha$ and $n$, by fitting it to the data and then relating by empirical correlation to basic soil properties (Vereecken et al., 1992; Wösten et al., 1995; Schaap et al., 1998; Minasny and McBratney, 2002; Rawls and Brakensiek, 1985; Van Genuchten et al., 1992; Wösten et al., 2001; Vereecken et al., 2010). One of the most commonly used water retention curves is the Van Genuchten model (1980). Schaap et al. (2001) developed the Rosetta package based on the artificial neural network method (ANN), which implements five hierarchical models to predict these parameters with well-defined limits (the soil texture classes only) and the most widespread input data (texture, density, and one or two points of water retention).

PTFs for point and parametric estimation of water retention curves from basic soil properties can be developed by using multiple regression methods (Lin et al., 1999; Mayr and Jarvis, 1999; Tomasella et al., 2000). An advantage of regression techniques is that most fundamental input parameters can be determined using stepwise regression.

Moreover, using pedotransfer functions in different environments either underestimate or overestimate the water retention. Several studies have shown that water retention is a complex function of soil structure and composition (Rawls et al., 1991; Wösten et al., 2001; Rawls et al., 2003). Applications of PTFs on different textural or structure classes may also be a source of uncertainty. Soil-water potential and hydraulic conductivity vary widely and non-linearly with water content for different soil textures. Experience has shown that soil texture predominantly determines the water-holding characteristics of most agricultural soils (Saxton et al., 1986). The relationship between the soil water retention curve (SWRC) and particle size distribution, was investigated in many studies (Jonasson et al., 1992; Minasny et al., 2006; Ghanbarian et al., 2009; Xu Yang et al., 2013; Tae-Kyu Lee et al., 2014). Williams et al. (1983) and Saxton et al. (1986) report that the PTFs depend mostly on texture, and other factors such as bulk density, structure, organic matter, clay type, and hysteresis may all have a secondary impact.

The variability of PTF response depends on variability and uncertainty of one or more input variables. The uncertainty analysis in the variety of pedotransfer (PTF) available approaches is a necessity to minimise the error in estimation and identify its source. Recently, sensitivity analysis techniques and uncertainty analysis have received considerable attention in the study of PTF (Nemes et al., 2006b; Kay et al., 1997; Grunwald et al., 2001; Deng et al., 2009; Moeys et al., 2012; Loosvelt et al., 2013). The question is, which variable input is the principal one or the better complimentary predictor of water retention, and at which potential? The global sensitivity analysis (GSA) allows us to study how the uncertainty in the output of a model can be apportioned to different sources of uncertainty in the model input (Saltelli et al., 2000). Generally, the GSA is very useful to know which variables mostly contribute to output variables (Jaques et al., 2004).

The objective of this study is:



1. Deriving and validating two approaches of PTFs from basic soil properties, using regression
methods:
• Point PTFs for estimation of soil water retention at -33 kPa and at -1500 kPa;
• Parametric PTFs for estimating the Van Genuchten curve parameters and comparing the
predictive performance with the Rosetta models;
2. Studying the impact of each input perturbation on the PTFs output.
**II.    Materials & methods**
1. The database
The PTFs are developed by using a database collected from some Algerian regions. Subset 1
containing 70 % of the samples (n = 189) from the coastal plain of Annaba located in the north-eastern
part of Algeria (n=13), the plain of Beni Slimane of Media (n=42), the Kherba El Abadia plain of Ain
defla (n=54) and samples randomly selected from Lower Cheliff plain in northwestern of Algeria (n=
80), soil series was used as the calibration set. Subset 2 with the remaining 21% (n = 53)
from  Benziane valley in the south west lower Cheliff plain, soil series was selected to verify the PTFs
(Table 1).  The depth of the two upper horizons varies from site to site with maximum of 30 cm for
surface horizons and upper than 30 cm for subsurface horizons.
The particle size analysis, conducted using the international Robinson's pipette method. Soil samples
taken by cylinders of 500-1000 cm3 (According to the case) were used to determine soil bulk density
(BD).The water retention values at -33 kPa and -1500 kPa were obtained by Richards's apparatus for
samples were collected in moisture nearby to field capacity, by cylinders with a volume of 100 cm3.
The water content measurements were conducted by gravimetric method at 105 C° (24h). The organic
carbon content was determined by wet oxidation method
The soil water retention model of Van Genuchten (1980) is defended as:

$$\theta(h) = \theta_r + \frac{\theta_s - \theta_r}{(1 + |\alpha h|^n)^m} \qquad\qquad (1)$$
Where $\theta_r$ and $\theta_s$ are the residual and saturated soil water contents (cm$^3$ cm$^{-3}$), respectively, and $\alpha$
(cm$^{-1}$) and n are the shape factors of the water retention function. The Van Genuchten parameters will
be calculated for each soil sample using the RETC code (Schaap et al., 2001). The "m" parameter
was calculated as follows: m = 1 -1 / n
2. Evaluation criteria
The PTF are regularly assessed by the comparison of the values that they predicted and the
measured values (Pachepsky and Rawls, 1999). To discuss the validity of PTF developed, we used
the following: the mean prediction error (ME) to inform the bias of the estimate; the mean square error
(RMSE) as an estimator of quality prediction that was frequently used in the literature on the
pedotransfer functions; and the index of agreement (d) developed by Willmott and Wicks (1980), and
Willmott (1981) as a standardised module in order to measure the model prediction error. They are
calculated using the following equation, respectively:



$$ME = \frac{1}{n} \sum_{i=1}^{n} (\theta_p - \theta_m)$$
(2)

with n, number of horizons, $\theta_p$, and the predicted volumetric water content and $\theta_m$ the measured
volumetric water content. The estimate is even less skewed than ME and is close to 0. When ME is
positive, PTF tested overestimated $\theta_m$ and when it is negative PTF tested underestimated θ.
$$RMSE = \left\{ \frac{1}{n} \sum_{i=1}^{n} (\theta_p - \theta_m)^2 \right\}^{\frac{1}{2}}$$
(3)

Thus, when the mean square error (RMSE) is low, the better the estimate.

$$d = 1 - \frac{\frac{1}{n} \sum_{i=1}^{n} (\theta_p - \theta_m)^2}{\sum_{i=1}^{n} \left[ \left| (\theta_p - \overline{\theta}_m) \right| + \left| (\theta_m - \overline{\theta}_m) \right| \right]^2}$$
(4)

The index of agreement (d) Varies between 0 and 1. Close to of 1 indicates a better matching between
measured and predicted values (Willmott and Wicks, 1980; Willmott, 1981).
3.    Global sensitivity analysis (GSA)
The global sensitivity analysis consists of determining which part of the variance of model response is
due to the variance of which input variable or group of inputs. These methods quantify the impact of
the parameters by the calculation of global sensitivity indices.

The Sobol method (Sobol, 1990) is an independent global sensitivity analysis that is based on

decomposition of the variance or may manage the functions and non-linear and non-monotonic
models. The Sobol model is represented by the following function:

$Y = f (X_1, X2, X3, ........, X_p)$        (5)

Where Y is the model output (or objective function) and X=(X1,…,Xp) is the variable set.
*a.    Sobol decomposition of variance*
The total variance of Y is defined; then:

V(Y) = V (E (Y |X)) + E (Var (Y |X))        (6)

When the input variables $X$i are independent, the variance decomposition of the model is:

$V(Y) = \sum_{i=1}^{p} V_i + \sum_i \sum_j V_{ij} + + \sum_i \sum_j \sum_p V_{ijp} + \dots\dots + V_{1,2,3,\dots p}$        (7)

$V_i = V [ E(Y|X_i)]$

$V_{ij} = V [ E(Y|X_i, X_j)] - V_i - V_j$

$V_{ijp} = V [ E(Y|X_i, X_j, X_p)] - V_{ij} - V_{ip} - V_{jp} - V_i - V_j - V_p$

Dividing $V_i$ by V(Y) we obtain the expression of the first-order sensitivity index noted $S_i$ such that:

$$S_i = \frac{V_i}{V(Y)} = \frac{V[E(Y/X_i)]}{V(Y)}$$        (8)

The term $S_i$ is the measure that guarantees an informed choice in the cases where the factors are
correlated and interact (Saltelli and Tarantola, 2002). This index is always between [0.1], and
represents a proper measure of the sensitivity used to classify the input variables in order of
importance (Saltelli and Tarantola, 2001).





To calculate the variation of sensitivity index ($V_{Si}$) we propose:

$V_{Si} = \left( \frac{V[E(Y/X)]}{V(Y)} - \frac{V[E(Y/Xi=Xi*)]}{V(Y)} \right) * 100$                    (9)
$V_{Si}$ > 0   and $S_i$ close to 1 indicate increasing accuracy of PTFs (+: favourable impact);
$V_{Si}$ < 0    and $S_i$ close to 1 indicate increasing accuracy of PTFs (+: favourable impact);
$V_{Si}$ > 0   and $S_i$ close to 0 indicate decreasing accuracy of PTFs (- : adverse impact);
$V_{Si}$ < 0   and $S_i$ close to 0 indicate decreasing accuracy of PTFs (- : adverse impact).
Moreover, coupling the RMSE and sensitivity index $S_i$ allowed us to detect the contribution of each
variable for the improvement of the quality of prediction of PTFs.
**III.     Results and discussion**

1.   The PTF derived

We chose to use the Rosetta PTFs in this study because it is one of the latest PTFs and gave
reasonable predictions in several evaluation studies (Nemes et al., 2003). The quality prediction of
point and parametric PTF developed in this study will be compared with the three Rosetta PTFs (H1,
H2, and H3). Three Rosetta models (H1, H2, and H3) were selected because they require the texture
and bulk density as inputs.

In Table 3, the majority of PTFs evaluated underestimate the soil water retentions except the

point model at the two pressure points (-33 kPa and -1500 kPa). The hierarchy Rosetta model H2,
which considers only texture as input, gave a smaller ME value compared with both H1 and H3
hierarchies models (- 0.0728; -0.0436 $cm^3 cm^{-3}$ at -33 kPa and -1500 kPa, respectively$^)$.

The poor ME values indicate better estimates of PTFs; they were produced after the

application of PTF points followed by the PTF parameters.

Among the five tested models in the Lower Cheliff soils, the PTF points (MLR) derived from a

database taken from some Algerian soils had the lowest values of RMSE (0,041 and 0,044 $cm^3 cm^{-3}$ at
-33 kPa and -1500 kPa, respectively). Performances equivalent or superior to PTFs derived by
multiple regression methods have been reported (Minasny et al., 1999; Nemes et al., 2003). However,
the non-linear models (parametric PTF) adapt better than the Rosetta models based on the artificial
neurons network (RMSE: 0.0613; 0.0605 $cm^3 cm^{-3}$ at -33 kPa and -1500 kPa, respectively).
Furthermore, the RMSE and the ME values of three Rosetta models show that H2 is better than H1
and H3 (Table 3).

In term of predictors, the results show that the OM improves the quality of adjustment.

Other evaluation criteria noted that the index of agreement also shows that the point PTF is

more suitable for Lower Cheliff soils than the parametric PTF (Fig. 6) with values *of (d)* (0.9975,
0.9911 $cm^3 cm^{-3}$). A similar comparison in different regions was made by Minasny et al. (1999),
Tomasella et al. (2003) and Ghorbani Dashtaki et al. (2010). All have reported similar
differences between these two types of PTFs.



While no significant difference in accuracy was found between the parametric PTFs and
Rosetta-H2 at -1500 kPa with RMSE value (0.0605 cm$^3$ cm$^{-3}$ and 0.0636 cm$^3$ cm$^{-3}$). The poor accuracy
of Rosetta PTFs can depend on several factors such as the similarity between the application region
and the database region source, geology, and bioclimatic context (Wagner et al., 2001; Wösten et al.,
2001; Ghorbani et al., 2011; Touil et al., 2016).
2.   Sensitivity index before the textural classification

In the development of pedotransfer functions, using the particle-size distribution (PSD) as
input is generally the common approach (texture as a global expression of the particle size
distribution, clay, silt and sand content), and its contribution is fundamental to understanding the
process of retaining water at different pressure points, although various physical and chemical
characteristics are used to describe the water retention curve such as the bulk density and organic
matter.
In this section, the importance of each input variable is assessed by the first order sensitivity
index ($S_i$). It is clear for the PTFs developed, the organic matter (OM %) and the clay percentages (C
%) are the variables that have the most impact particularly on point PTF (MLR) estimation in two
pressure points with $S_i$ in order to (OM: 0.821; 0.630) and (C %: 0.782; 0.585) at -33 kPa and -1500
kPa, respectively (Fig. 2). The percentage of silt (Si %) is in a second range of importance in
parametric PTF (0.576 at -33 kPa) after OM, followed by the bulk density and clay (Fig. 2). The
$S_i$ values class the sand content in third order in MLR (0.262; 0.162), thus its impact on the
parametric model is almost insignificant with very low values index ($S_i$: 0.077; 0.017) at -33 kPa and -
1500 kPa, respectively.
The prediction quality of point PTF developed by linear multiple regression (MLR) can be
explained first by taking into account the basic characteristics of soil as an input through the texture
and structure information given by the bulk density. Secondly, the MLR formation is mainly based on
these input variables compared to parametric PTF using the non-linear multiple regression (MNLR),
which has as input more than texture and the bulk density, but also other parameters (parameters of
the Van Genuchten curve: $\Theta_r$, $\Theta_s$, α, n, m).

3.   Sensitivity and uncertainty analysis after the textural classification

This section will analyse the sensitivity of multiple regression methods (linear and non-linear) developed
for the basic soil characteristics on estimating water retention in different textural classes. Water
retention and conductivity are directly related to the geometry of the pore network, depending on the
size and assemblage of the elementary soil particles. In this order we have grouped the samples
into three classes of particles according to textural classes of the FOA guidelines (FOA, 1990),
which give a very fine class (N = 12), fine (N = 31) and medium (N = 10).
The results show the improvement of the quality estimation of PTFs after textural stratification,
particularly in medium class (Fig. 4). Indeed, a better prediction was recorded by point PTF (RMSE
= 0.027 cm$^3$ cm$^{-3}$) and parametric PTF (RMSE = 0.038 cm$^3$ cm$^{-3}$) at -1500 kPa. The stability in



estimation of PTF before and after classification is noted in the very fine class (Fig. 4). This can be
explained by the difficulties there are for linking the water retention properties of the samples with
their size distribution as their structural state may be variable.
**1. Sand content:** After textural classification, PTFs developed (MLR and MNLR) are always
sensitive primarily to the sand fraction in fine and medium classes (Table 5). The variation of the
first sensitivity index in point PTF is significantly greater in the medium texture class at the two
pressure points (-33 kPa and -1500 kPa). Into the MNLR, sand has the most influence particularly
when it is applied to the fine class (-40.9%, 18.9% at -33 kPa and 1500 kPa) and  medium class (-
16.7% at -1500 kPa).

The sensitivity index of a variable quantifies the influence of its uncertainty on the output. This

is the part of the variability output explained by the variability input. What has been confirmed after
calculating the variation of the first order sensitivity index ($V_{Si}$), is that the PTFs developed are still very
influenced by the variability of sand at -33 kPa more than at -1500 kPa. This impact can be explained
by the irregularity of the dispersion of sand content in the validation database with a coefficient of
variation (CV) approximately 119% compared to the other input variables (33%; 18%; 9%; 57% for
clay, silt, bulk density, organic matter, respectively). This heterogeneity of the sand data series clearly
influences the uncertainty of pedotransfer functions response.

Moreover, looking at the matrix correlation (Table 6), the clay and silt fraction are significantly

correlated with the sand content. Saltelli and Tarantola (2002) observe that when $X_1$ and $X_2$ are
correlated with a third factor $X_3$, the sensitivity index calculated depends on the force of this correlation
as well as the distribution of $X_3$. In this case, the index power may be influenced by this statistical
association, as we can explain the higher value difference of index variation of sand percentage
compared with the other variables (Fig. 2).

We can see that point PTF (MLR) produces a lower error of estimation when the variation

sensitivity index calculated for sand is the leading [MLR in the medium class: RMSE (0.030; 0.027 cm$^3$
cm$^{-3}$) with $V_{Si}$ = (-103%. 86.4%) at -33 kPa and -1500 kPa, respectively]. A negative $V_{Si}$ of sand
content when the latter is fixed is noticed on all texture classes (Table 5). This can be explained by the
proportional relationship between the sand and clay content, particularly in the dataset of validation
with dominant clay texture. Insignificant sensitivity of sand was recorded in very fine texture. Rawls et
al. (2003) observed that 10% of sand provides an increase in water retention at low clay content and a
decrease in water retention at high clay content of more than 50%.

It is important to note the relationship of the Van Genuchten water retention curve parameters

(especially n and α) and particle size distribution were conducted recently in many studies (e.g.
Minasny et al., 2007; Benson et al., 2014) in order to explain why the sand impact increase in fine
texture class in parametric PTF. It can be explained by the majority presence of sand and clay content
as input on parametric PTFs. For soils with clay content between 35% and 70%, water content is
highly influenced by the percentage of sand in the soil (Loosvelt et al., 2013).

Moreover, when the sand content of the sample increases to 60%, the drying rate is quicker

and water absorbing ability is weaker compared with the small sand content. When the sand content





decreases to 20% the small pores occupy a large part of the pore structure, making the soil compact
(Hao et al., 2015).
**2. Bulk density**: the second most influential variable on the point PTF (MLR) response is by variation
of sensitivity index on all textural class, mainly in the very fine textural class with elevated values at -
33 kPa ($V_{Si}$ = -50, 5%). In the parametric PTF, the bulk density influences the medium class at -33
kPa. The result shows that the favourable impact of BD is according to the accuracy of quality
estimation at -33 kPa in the medium class on two developed approaches of PTFs (Table 5). The very
fine textural class represents 16 surface samples (0–30 cm) with a dominance of clay texture. In a
similar study on clay soils, the volumetric water content is hugely related to the inverse of bulk density
at field capacity (Bruand et al., 1996). It may also explain the fact that many soils with high clay
content in the database are vertisols in which they increase in organic matter, decrease bulk density
and decrease the volumetric water content (Rawls et al., 2003). The inclusion of the bulk density in the
development PTFs leads to a pore volume that cannot retain water for the potential range studied (-33
kPa and -1500 kPa).
In addition, the soil structural information characterised by measurements of bulk density is an
indirect measurement of pore space and is affected primarily by texture and structure. For structure-
less soils, primarily coarse and medium textured soils, the capillary pore-size distribution can be
satisfactorily described by particle size distribution. The medium texture relates in a general way to the
pore-size distribution, as large particles give rise to large pores between them, and therefore, is a
major influence on the soil water retention curve (Arya and Paris, 1981; Nimmo, 2004)**.** With this
variable, and the texture used as inputs in point PTF (MLR), the nearest experimental results are
obtained. The results of this study can confirm that the effect of the use of the soil structural
information on the estimation of the soil water retention depends on the use of regression techniques
(Nguyen et al., 2015).

**3. Clay content:** For medium texture the favourable sensitivity is determined by clay content at -33
kPa. This can be explained by the reduction of clay percentage in the medium class (mean of clay (%)
= 23%), which produces fewer errors at -33 kPa. The highest impact of clay (%) was observed at -
1500 kPa on the point and parametric PTF in different textural classes (Fig. 4). The clay content of
soils is a major predictor for modelling the permanent wilting point of soils (Minasny et al., 1999).
Moreover, in this study, the accuracy of PTFs decrease when they were applied to some soil
samples with the Clay (%) > 60% (Fig. 4). In the very fine class, insignificant sensitivity is recorded at
all pressures defined in this study. In this class, the variation of clay is much lower, for the reason that
the latter is only the dominant solid fraction, and this can explain the smaller variation of sensitivity
index after fixing the clay percentage. The very fine and fine classes have advanced water retention t
more those of the medium class, because it quickly drains water initially retained. In other words, the
use of clay as PTF input in medium textured soils leads to results that are more or less erroneous.
**4. Silt content:** In this study silt was introduced as an explanatory variable only in MNLR parametric
PTFs. This fraction is known for its ability to retain water to high and medium potential. The analysis
results show that the global sensitivity class and the silt as the second input were the most disturbing



estimates of water retention at -1500 kPa more than at -33 kPa on MNLR model. After textural
stratification, the main values of $V_{Si}$ have been found in medium class (-36.7% to -1500 kPa). The
lowest were recorded in the very fine class, or the texture is pure clay. It is clear that the percentage of
silt has a very important role in estimating of the Van Genuchten parameters ($\alpha$, n), and consequently,
its use as input in MNLR influences the estimate in the medium and fine class. Nevertheless, there is
a favourable impact recorded in fine class at -1500 kPa. Its presence with the clay content in the fine
class has led to a better pedological interpretation of the soil water retention in silty-clay texture. The
plant-available water content variation is more related to sand and silt than to clay content (Reichert et
al., 2009).
**5. Organic matter content:** The most insignificant variation of sensitivity index ($V_{Si}$) after textural
stratification is attributed to the organic matter content. This can be explained firstly by the poor OM on
the Algerian soils. Lal (1979) and Danalatos et al. (1994) did not find any effect of organic matter
content on water retention; the latter attributed it to the generally low organic matter content in their
samples. Secondly, homogeneity of the data for OM content in every textural class decreases the
variation of PTFs response, as the latter is always considered the better predictor of soil water
retention particularly in clayey soils. However, positive sensitivity impact is observed on parametric
PTF in medium-textured soils at -33 kPa where the OM is used as input to predict saturated soil water
contents. Soil water retention at -33 kPa is affected more strongly by the organic carbon than at -1500
kPa (Rawls et al., 2003). The sensitivity analysis made by Rawls et al. (2003), in order to study the
role of organic matter content as predictor, shows that water retention of coarse-textured soils is much
more sensitive to changes in organic carbon as compared with fine-textured soils. Bauer and Black
(1981) found that the effect of organic carbon on water retention in disturbed samples was substantial
in sandy soil and marginal in medium and fine textured soils.
**IV. Conclusion**
The present study suggests that the soil water retention is controlled by different variables such as
organic matter, clay content and sand content at different points of soil water potential, and not directly
related to the parameters of the water retention curve such as the van Genuchten model. The
reliability tests show that the point PTF predicts more accuracy than the parametric models. Indeed,
the derived parametric PTFs provide better estimates than the Rosetta models that were originally
developed from a large intercontinental database.
Furthermore, the global sensitivity analyses show that the mathematical formalism of PTF
models and their input variables react differently in terms of point pressure and textural class:
• It is clear that the soil water retention is highly related to the OM and clay content.
• After textural classification, the two approaches of PTFs developed (MLR and MNLR) are
always sensitive primarily to the sand fraction in fine and medium class at -33 kPa more
than at -1500 kPa.
• The results show that the favourable impact of BD is according to the accuracy of quality
estimation of the two approaches of PTFs developed at -33 kPa in the medium class.



• The accuracy of PTFs decrease when they were applied to some soil samples with the
Clay (%) > 60%.
• The most insignificant variation of sensitivity index $V_{Si}$ after textural stratification is
attributed to the organic matter content in Algerian soils.

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

**Tables:**
**Table 1**. Soil characteristics used for the development and validation.

| | Granulometry | | | | | VWC ($cm^3$ $cm^{-3}$) | |
|---|---|---|---|---|---|---|---|
| | S (%) | Si (%) | C (%) | BD (g/$cm^3$) | OM (%) | - 33 kPa | -1500 kPa |
| ***Samples used for Deriving PTF (n = 189)*** | | | | | | | |
| Average | 17.81 | 39,23 | 42.97 | 1.71 | 0.95 | 0.44 | 0.27 |
| Standard Deviation | 10.32 | 10.76 | 13.90 | 0.20 | 0. 93 | 0.09 | 0.08 |
| Min | 1.00 | 9.20 | 4.00 | 0.60 | 0.08 | 0.13 | 0.03 |
| Max | 50.00 | 67.00 | 84.30 | 2.10 | 8.40 | 0.73 | 0.56 |
| Coefficient of Variation (CV) | 0.58 | 0.27 | 0.32 | 0.12 | 0.98 | 0.21 | 0.31 |
| ***Samples used for testing the PTF (n = 53)*** | | | | | | | |
| Average | 12.50 | 41.58 | 45.92 | 1.49 | 0.87 | 0.40 | 0.21 |
| Standard Deviation | 14.84 | 7.62 | 14.94 | 0.13 | 0.50 | 0.10 | 0.07 |
| Min | - | 29.00 | 9.00 | 1.15 | 0.20 | 0.14 | 0.07 |
| Max | 59.00 | 58.00 | 70.00 | 1.73 | 2.74 | 0.57 | 0.45 |
| Coefficient of Variation (CV) | 1.19 | 0.18 | 0.33 | 0.09 | 0.57 | 0.24 | 0.35 |

*(\*) S: sand, C: clay, Si: silt, BD: bulk density, OM: organic matter, VWC: volumetric water content*
*CV: coefficient of variation.*
**Table 2**. Multiple regression coefficient $R^2$ and regression coefficients of models developed.

| | | Points PTF (MLR) | | | Parametric PTFs (MNLR : Cubic Model) | | |
|---|---|---|---|---|---|---|---|
| | | -33 kPa | -1500 kPa | $\Theta s$($cm^3$ $cm^3$) | $\Theta r$($cm^3$ $cm^3$) | α | n |
| | *Inputs (\*)* | *S%, C%, BD, OM* | *S%, C%, BD,* | *S%, C%, BD, OM* | *C%,S%* | *Si%, S%* | *C%, Si%* |
| | $R^2$ *multiple* | 0.74 | 0.66 | 0.62 | 0.67 | 0.60 | 0.66 |
| | *a* | 0.0246 | -0.0627 | 0.4136 | $9.00 \times 10^{-02}$ | $3.00 \times 10^{-03}$ | 2.90 |
| | *b* | -0.0040 | -0.0029 | -0.0013 | $7.78 \times 10^{-04}$ | $-1.00 \times 10^{-04}$ | $-2.77 \times 10^{-03}$ |
| | *c* | 0.0012 | 0.00165 | 0.0002 | $3.20 \times 10^{-04}$ | $8.90 \times 10^{-05}$ | $-9.48 \times 10^{-02}$ |
| | *d* | 0.2554 | 0.1837 | 0.0177 | $-6.36 \times 10^{-05}$ | $5.40 \times 10^{-06}$ | $-3.66 \times 10^{-04}$ |
| ***Regression Coefficients*** | *e* | 0.0067 | | - 0.0018 | $1.20 \times 10^{-05}$ | $-4.50 \times 10^{-06}$ | $2.03 \times 10^{-03}$ |
| | *f* | - | - | - | $9.30 \times 10^{-07}$ | $-7.30 \times 10^{-08}$ | $2.49 \times 10^{-06}$ |
| | *g* | - | - | - | $-1.00 \times 10^{-07}$ | $4.50 \times 10^{-08}$ | $-1.50 \times 10^{-05}$ |
| | *h* | - | - | - | $9.00 \times 10^{-02}$ | $7.70 \times 10^{-06}$ | $2.84 \times 10^{-04}$ |
| | *i* | - | - | - | $7.78 \times 10^{-04}$ | $-3.10 \times 10^{-08}$ | $4.91 \times 10^{-06}$ |
| | *j* | - | -- | - | $3.20 \times 10^{-04}$ | $-3.10 \times 10^{-08}$ | $-5.32 \times 10^{-06}$ |

*(\*) S: sand, C: clay, Si: silt, BD: bulk density, OM: organic matter (respectively)*



**Table 3.** Evaluation criteria of PTFs at -33 kPa and -1500 kPa.

|  |  |  | *-33 kPa* | *-1500 kPa* |
|---|---|---|---|---|
| *ME (cm³ cm⁻³)* | *PTF Point* | MLR | 0.0188 | 0.0261 |
|  | *Parametric PTF* | MNLR | -0,0016 | -0.0020 |
|  | *Rosetta* | H1 | - 0.0902 | -0.0458 |
|  |  | H2 | - 0.0728 | -0.0436 |
|  |  | H3 | -0.0991 | -0.0552 |
| *RMSE (cm³ cm⁻³)* | *PTF Point* | MLR | 0.0414 | 0.0444 |
|  | *Parametric PTF* | MNLR | 0.0613 | 0.0605 |
|  | *Rosetta* | H1 | 0.1170 | 0.0738 |
|  |  | H2 | 0.0970 | 0.0636 |
|  |  | H3 | 0.1280 | 0.0749 |
| *d (cm³ cm⁻³)* | *PTF Point* | MLR | 0.9975 | 0.9911 |
|  | *Parametric PTF* | MNLR | 0.9938 | 0.9775 |
|  | *Rosetta* | H1 | 0.9623 | 0.9427 |
|  |  | H2 | 0.9775 | 0.9597 |
|  |  | H3 | 0.9519 | 0.9331 |

**Table 5**. The variation of first order sensitivity index in the textural classes.

|  |  | Tex-class | Si (%) $V_{Si}$ | A.E | S (%) $V_{Si}$ | A.E | C (%) $V_{Si}$ | A.E | BD (g/cm³) $V_{Si}$ | A.E | OM (%) $V_{Si}$ | A.E |
|---|---|---|---|---|---|---|---|---|---|---|---|---|
| **RML** | **at -33 kPa** | VF | *Abs* |  | -1.2 |  | -0.4 |  | -50.5 | - | 4.6 |  |
|  |  | F | *Abs* |  | -43.2 | - | -10.7 | - | -39.9 | - | 0.2 |  |
|  |  | M | *Abs* |  | -103.3 | - | -27.5 | + | -44.4 | + | -5.7 |  |
|  | **at -1500 kPa** | VF | *Abs* |  | -0.3 |  | 0.9 |  | -27.3 | - | 1.1 |  |
|  |  | F | *Abs* |  | -46.2 | - | -20.7 | - | -41.6 | - | 0.1 |  |
|  |  | M | *Abs* |  | -86.4 | - | -52.9 | - | -22.9 | - | -2.3 |  |
| **MNLR** | **at -33 kPa** | VF | 0.4 |  | -0.2 |  | 0.1 |  | -00.1 |  | -0.05 |  |
|  |  | F | -1.6 |  | -40.9 | - | -1.1 |  | -2.5 |  | -0.1 |  |
|  |  | M | 15.0 |  | -5.2 |  | 15.1 | + | 21.6 | + | 22.3 | + |
|  | **at -1500 kPa** | VF | - 4.6 |  | -0.3 |  | -1.8 |  | -1.4 |  | -00.5 |  |
|  |  | F | 28.6 | + | 18.9 | - | 4.6 |  | 0.4 |  | 0.1 |  |
|  |  | M | -36.7 | - | -16.7 | - | -22.6 | - | 8.9 |  | -8.4 |  |

$V_{Si}$: variation first sensitivity index;     A.E.: improving estimation; +: favourable impact; - : adverse
impact.
**Table 6**. Correlation matrix (coefficient of Pearson) of validation database (n=53).

| Variables | Si% | S % | C (%) | BD (g/cm³) | OM (%) |
|---|---|---|---|---|---|
| S% | **1** |  |  |  |  |
| S % | **-0.334** | **1** |  |  |  |
| C % | -0.159 | **-0.878** | 1 |  |  |
| BD (g/cm3) | 0.164 | -0.185 | 0.11 | **1** |  |
| OM (g/100g) | -0.174 | -0.166 | 0.263 | -0.19 | **1** |

The values which are in bold are different from 0 to a level of signification
α = 0.05




**Figures:**
**Figure 1**. Scatter plots of measured versus predicted soil water retention by H2 Rosetta.

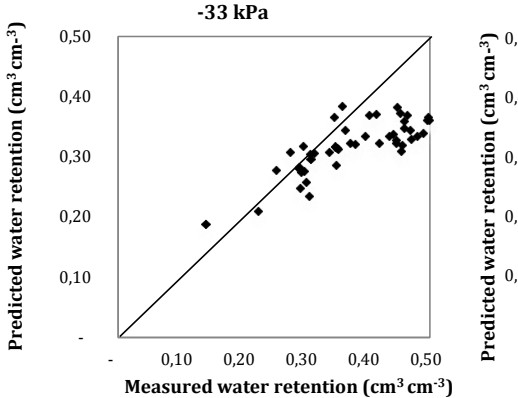
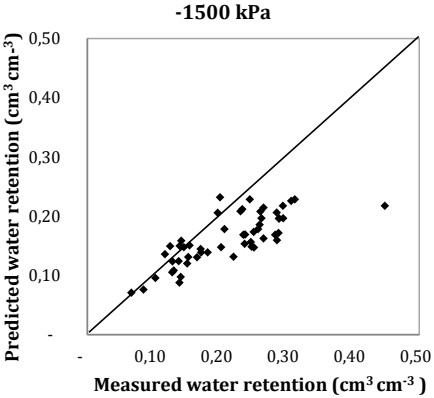

**Figure 2.** First order sensitivity index

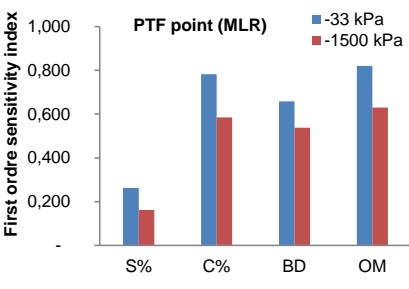
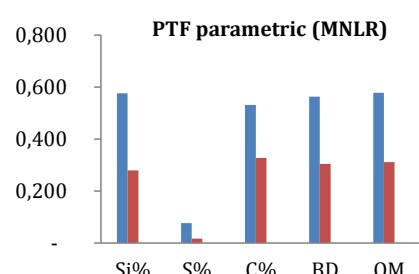

**Figure 3**. Textural triangle proposed by FOA (1990).

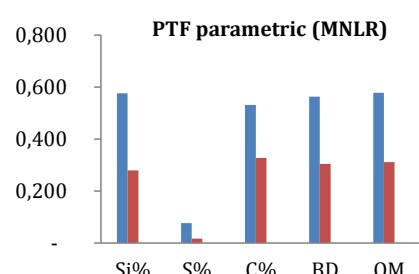

**Figure 4.** The RMSE criteria calculated with textural classification.




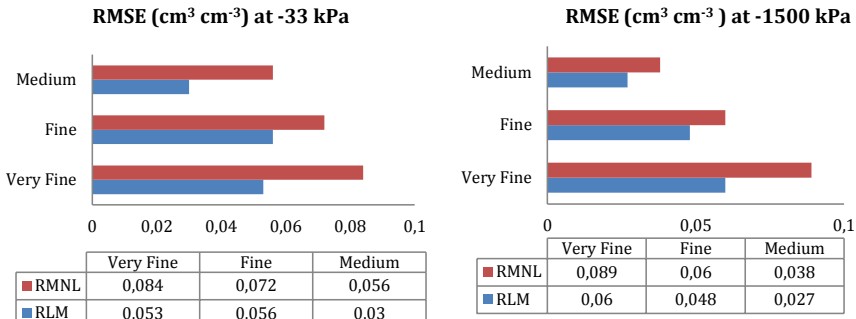

**Figure 5.** Variation of first sensitivity index with RMSE criteria after textural classification.

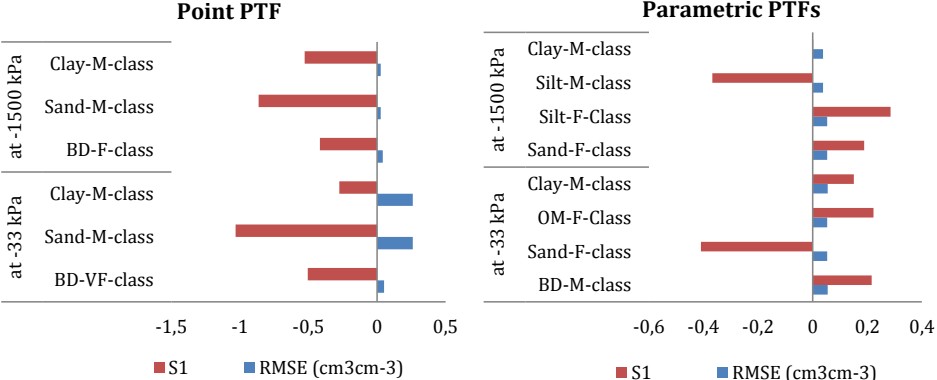

**Figure 6.** Scatter plots of measured soil water retention versus predicted soil water retention.

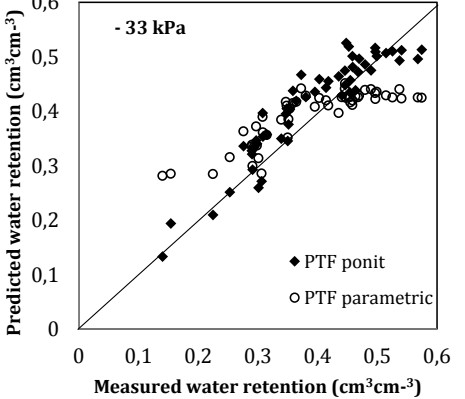
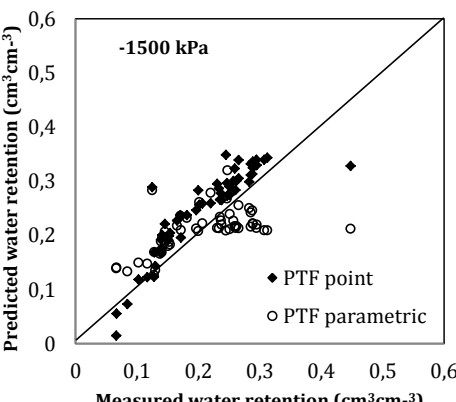
