# Peer review of "Sensitivity analysis of point and parametric pedotransfer functions for estimating water retention of soils in Algeria"

_SOIL, 2016_

## Referee Comment (RC1) · S. Barontini (Referee) · 4 Jul 2016

**1   General comments**

The article proposed by Touil and coauthors addresses an important issue, when pedotransfer functions (PTFs) are used to characterize the soil water retention curve (SWRC), that is the sensitivity of the predicted soil water contents to the predictor variables. The goal is to provide an insight on which are the most reliable predictors, among the most common ones (bulk density, grain size distribution, organic matter content).

[Figure]

In order to do so, the Authors investigated a sample of more than two hundred soils with poor organic matter content, splitted into two subsamples, one for interpolation (189 soil samples) and one for validation (53 soil samples), and compared the obtained results with those predicted by means of Rosetta database.

PTFs were approached both point by point and in parametric form. Multiple linear regression (MLR) was used to find a relationship between soil water contents and the explanatory variables, and the multiple nonlinear regression (MNLR) was used to find a relationship between the parameters of the van Genuchten (1980) SWRC and the explanatory variables.

Then the global sensitivity analyses (GSA) method was used to perform the sensitivity analysis of the predicted variables against the explanatory ones.

The database is rich, the methodology and the results can be interesting, so that I recommend this article for publication in Soil provided that the Authors fix some issues of major importance and some structural aspects of the presentation.

My major concerns are about the parametric approach to PTFs.

In lines 106 to 113 it is said that the RETC code was used to fit the van Genuchten SWRC to the experimental data, but at line 102 it is reported that the experimental data were determined at two tensiometer–pressure potentials, that are -33 and -1500 kPa. Therefore it seems that the four parameters of the SWRCs ($\theta_r$, $\theta_s$, $\alpha$, $n$, while $m$ is constrained to $m = 1 - 1/n$) are fitted by means of two experimetal points only for, each curve.

If this is the case, and no other constraints were introduced, the set of parameters is not univocally identified for each soil, and the further analyses on the parametric approach loose their significance.

I therefore recommend that either (1) the Authors better detail the followed procedure for this approach, so that it is clear how main experimental points the procedure is

based on or whether there were other constraints to univocally identify the fitted parameter set; or (2) they remove the part about the parametric approach and better develop that about the point approach.

Moreover (3) I encourage the Authors to explicitly present the PTFs they obtained for the investigated sample of soils.

In the following lines, detailed comments and some technical notes will be provided.

**2  Detailed comments and techincal notes**

ll.37—38 Explain whether it refers to the hydrological state of the soil or to the characterization of the hydrological properties;

l.47 $\Theta \to \theta$ (in all the paper);

l.51 Uniform all the paper to the version "van Genuchten" (or to "Van Genuchten");

l.59 "different environments from which they were derived for"

l.63 "and hydraulic conductivity as well";

ll.93—96 Check the percentages, or probably better explain the consistency of the whole database;

l.103 "moisture" -> "water content". Field capacity or soil saturation? Samples in Richards apparatus are usually saturated. Moreover field capacity (regarded to as the soil water content which remains in the soil after abundand imbibition and when percolation is materially decreased) can be quite a small water content, even smaller than the water content at 33 kPa;

l.106 "defended"->"defined";

l.119 Add something like "the following measures of the errors", or something else, to make the article more readable;

l.131 Check equation (4), I think that there should not be $\frac{1}{n}$;

l.145 Title not necessary;

l.161 Explicit what does the constraint $X_{i\star}$ stand for;

l.177 Table 2 is not cited before Table 3. This is a good point to explicitly provide the formulae of the obtained PTFs;

ll.199—203 I agree with this sentence, but in this case it can also be due to the undetermination of the interpolated parameters (see the General Comments);

l.233 Avoid referring to the conductivity as the framework of the article seems to be based on Mualem's predictive approach to the relative conductivity function (as it follows from the constraint on $m$);

l.244 and followings Consider the idea of collecting all the analyses regarding the texture in one paragraph only, thus restructuring the paragraphs regarding sand, sil and clay. This can strongly help the readability of the discussion. Many analyses of previous Authors are reported: I suggest to explicitly detail whether your results are according or discording to previous ones;

l.291 "They increase in organic matter" with. . . ?

l.306 and followings Typically clay is very important at characterising the water retention, even if it can loose sensitivity for great values of clay content: in which sense does it sound the statement of line 317?

l.353 I agree with the conclusion but it seems to be quite in contrast to what observed after the reported analyses and the last conclusion: I suggest to better detail this point or remove it.

Further minor comments: (i) correct some typos, (ii) check the consistency of the references list and alphabetically order it, (iii) change the colour of histograms and bar–graphs to ensure the readibility also in B&W printing.

―――――――――――――――

---

## Referee Comment (RC2) · Anonymous Referee #2 · 9 Aug 2016

Overall opinion

This is an interesting paper covering an important topic, namely the prediction of soil hydraulic properties, particularly the soil water retention curve for soils in Algeria. However, there are a number of issues that need to be addressed before publication could be recommended.

General comments

The real major issue that I have with this paper is its lack of novelty. A large number of papers on pedotransfer functions are being submitted to various peer-reviewed journals, which basically all follow the same pattern as this paper does:

[Figure]

1. Data are collected locally 2. 'Foreign' PTFs are tested 3. 'Home' PTFs are often developed, but not always 4. Home PTFs are deemed better - or a foreign PTF is found better than others.

I agree that using global sensitivity analysis is very useful in decomposing the variance of the response (soil water retention) into contributions from the individual input variables. However, this analysis does not add any new information on what is already known in literature about the contribution of various predictors to the predictive quality of point and parameter-based PTFs. This issue is long known and has been shown/commented on by many papers by now.

Another weakness of the paper is its lack of clarity in many parts of the text. I expanded more on this in the specific and technical comments and the authors need to work on that. Good proofreading and editing would considerably improve the quality of the manuscript.

Specific comments

Title: I would suggest: 'Sensitivity analysis of point and parametric pedotransfer functions for estimating water retention of soils in Algeria'.

Page 1, line 27: What did the authors mean by: 'favourable impact'?

Page 1, lines 39-40: I would suggest: 'hydrologists face the situation where soil hydraulic data such as water retention or hydraulic conductivity are often missing. Therefore, pedotransfer functions (PTFs) are used as an alternative to estimate these properties.'

Page 1, lines 40-41: I do not agree that reports on the evaluation of PTFs outside the area of development are rare (see general comments above). This is one of the main topics in PTF studies.

Page 2, line 54: Water retention points are not part of the widespread input data for PTF

Page 2, lines 57-58: I am missing something here; why should we call it an advantage?

Page 2, line 61: I would expect more recent references

Page 2, line 62: Could the authors provide some references?

Page 2, line 63: I would expect: 'Soil-water retention and hydraulic conductivity vary widely and non-linearly with soil water potential'

Page 2, line 68: Could the authors also provide more recent references

Page 3, lines 85-86: 'comparing the predictive performance with the Rosetta models': this looks like a third objective.

Page 3, line 88: I am missing a short description of the study area and information on soil types

Page 3, line 93: n has been used to design three different variables: (1) number of soil samples in a subset (Page 3, line 93); (2) shape factor of the water retention function (Page 3, line 111); (3) number of horizons (Page 4, line 125)

Page 3, line 96: What did the authors mean by: 'soil series was used as the calibration set'? See also Page 3, lines 97-98.

Page 3, lines 100-105: The authors should provide references for all the lab methods.

Page 3, lines 102-103: I would rephrase it as follows: 'The water retention values at -33 kPa and -1500 kPa were obtained by the Richards's apparatus. Undisturbed soil samples were collected near field capacity with 100 cm3-cylinders'

Page 3, line 122: What did the authors mean by: 'standardised module'?

Page 4, line 126: What did the authors mean by: 'The estimate is even less skewed than ME and is close to 0'?

Page 4, lines 141-142: '...may manage the functions and non-linear and non-monotonic models': this sentence is not clear to me. Please rephrase

Page 5, line 172: The first reason mentioned by the authors for selecting the Rosetta PTFs in their study seems weak to me as these PTFs have been published 15 years ago

Page 5, line 175: The authors should give more details on Rosetta models H1, H2 and H3

Page 5, lines 175-176: The second reason for selecting the Rosetta PTFs should be better explained

Page 5, line 188: What did the authors mean by 'adapt better'?

Page 5, line 193: 'Other evaluation criteria noted that the index of agreement also shows that the point PTF is...': this sentence is not clear and should be rephrased

Page 6, lines 199-200: Did the authors perform a significance test to confirm this?

Page 6, lines 216-217: '...with Si in order to (OM: 0.821; 0.630) and (C %: 0.782; 0.585) at -33 kPa and -1500 kPa, respectively (Fig. 2)': this sentence should be rephrased

Page 6, line 227: m is directly linked to n by a simple relation (see Page 3, line 113). Therefore, we should only consider 4 parameters: $\theta$s, $\theta$r, $\alpha$ and n

Page 7, line 241: What did the authors mean by 'The stability in estimation of PTF before and after classification'?

Page 7, lines 265-266: What did the authors mean by: 'when the variation sensitivity index calculated for sand is the leading'? Please rephrase

Page 7, line 276: What did the authors mean by 'the majority presence'? Please rephrase

Page 8, lines 290-294: These 2 sentences are not clear to me. Please rephrase

Page 8, lines 300-301: Which variable do the authors refer to?

Page 8, line 306: What did the authors mean by: 'favourable sensitivity'? Please rephrase

Page 8, line 315: What did the authors mean by: 'advanced water retention'? Please rephrase

Page 8, line 320 to Page 9, line 321: This sentence is not clear to me. How the global sensitivity class and the silt can be estimates of water content? Please rephrase

Page 9, line 322: What did the authors mean by: 'the main values of VSi'? Please rephrase

Page 9, line 323: 'or the texture is pure clay': I do not understand this part of the sentence. Please rephrase

Page 9, line 325: Do the authors refer to the estimate of water retention or of the van Genucthen parameters?

Page 9, lines 325-327: These 2 sentences are not clear to me. What did the authors mean by 'favourable impact', 'a better pedological interpretation'? Please rephrase

Page 9, lines 335-336: '...as the latter is always considered as the best predictor of soil water retention particularly in clayey soils'. Could the authors support this affirmation by references to recent literature?

Page 9, line 336: What did the authors mean by 'positive sensitivity impact'? Please rephrase

Page 9, lines 345-347: This sentence is very weak and does not add on what is already well known from previous studies

Page 9, line 348: '...predicts more accuracy than...': Please rephrase.

Page 9, lines 357-358: As a conclusive sentence, this is not clear to me. Please rephrase

Tables and figures - Tables and figures should be self-explanatory in their titles and contents. The authors should provide all the necessary information such as explanation for abbreviations, measurement units, etc. Please see also specific comments

- The order of captions of tables and figures should generally correspond with the order of appearance in the text

- What do the authors mean by cubic model in Table 2?

Technical comments

Title: I would suggest: Sensitivity analysis of point and parametric pedotransfer functions for estimating water retention of soils in Algeria.

Page 1, lines 20-21: the way values of RMSE are reported in the abstract is confusing. Page 1, line 21: RMSE values are in cm3 cm-3 as units for water retention. Page 1, line 28: medium textural class Page 2, lines 44-45: 'estimated' instead of 'constructed' Page 2, line 47: the parameters of the van Genuchten model ($\theta$s, $\theta$r, $\alpha$ and n) have been introduced before the van Genuchten model itself (line 51). It would be more logical to remove them in line 47 and insert them in line 53. Page 2, line 53: I would suggest: '...to predict the van Genuchten parameters ($\theta$s, $\theta$r, $\alpha$ and n) with soil texture classes only...' Page 2, line 54: 'bulk density' instead of 'density' Page 2, line 55: 'Pedotransfer functions' instead of 'PTFs' Page 2, line 60: 'water retention' instead of 'the water retention' Page 2, line 72: 'pedotransfer' should be deleted Page 2, line 76: 'complementary' instead of 'complimentary' Page 3, lines 82-83: I would suggest: 'Deriving and validating two approaches of PTFs using regression methods:' Page 3, line 87: 'input perturbation' is not appropriate. Please rephrase. Page 3, line 92: I would suggest: 'The PTFs are developed using a database of soil samples collected from some regions in Algeria' Page 3, line 93: 'contains' instead of 'containing' Page 3, line 99: 'more than 30 cm' instead of 'upper than 30 cm' Page 3, line 100: 'was conducted using...' Page 3, line 100: I would suggest: 'Undisturbed soil samples taken...' Page 3, line 101: '(According to the case)' should be deleted Page 3, line 104: I would suggest: 'Water content measurements were conducted by the gravimetric method' Page 3, line 106: The word 'defended' is inappropriate. Please rephrase Page 3, lines 111-112: 'were calculated' instead of 'will be calculated' Page 3, line 117: 'by comparing the values that they predicted' Page 3, lines 118-119: I would suggest: 'To discuss the validity of the PTF developed, we used the following criteria:' Page 3, lines 119-120: 'the root mean square error (RMSE)' instead of 'the mean square error (RMSE)' Page 3, line 120: 'of the quality of the prediction' instead of 'of quality prediction' Page 4, line 129: 'the root mean square error (RMSE)' instead of 'the mean square error (RMSE)' Page 4, lines 143-144: Is it X1 or X1? This needs to be uniform. Page 4, line 144: I would suggest: '...X=(X1,..., Xp) is the input variable set' Page 4, lines 147-149: The elements of equations 6 and 7 are not explained Page 4, line 157: 'between 0 and 1' instead of 'between [0.1]' Page 5, line 161: What is Xi* in equation 9? Page 5, line 168: I am missing sentences that introduce Table 1 and Table 2. Moreover, Table 2 is not mentioned in the whole text. Page 5, line 170: I would suggest as title: 'Development of PTFs' Page 5, lines 172-173: With respect to the title in line 170, this sentence should be placed at the end of the paragraph Page 5, line 177: 'soil water retention' instead of 'the soil water retentions' Page 5, line 184: I would suggest: 'the point MLR PTFs' instead of 'the PTF points (MLR)' Page 5, line 185: '0.041 and 0.044' instead of '0,041 and 0,044' Page 5, line 188: 'parametric PTFs' instead of 'parametric PTF' Page 5, line 189: 'neural' instead of 'neuron' Page 5, line 189: 0.0613 and 0.0605 cm3 cm-3 Page 6, line 199: 'while' should be deleted Page 6, line 207: 'PTFs' instead of 'pedotransfer functions' Page 6, line 209: 'fundamental to understand the...' instead of 'fundamental to understanding the...' Page 6, line 211: 'as bulk density' instead of 'as the bulk density' Page 6, lines 214-217: This sentence is too long and should be divided into 2 sentences. Page 6, lines 215-216: 'at two pressure points' instead of 'in two pressure points' Page 6, line 219: 'in third place' instead of 'in third order' Page 6, line 222: 'MLR' instead of 'linear multiple regression' Page 6, line 224: I would suggest: 'point PTF using MLR is mainly based on...' Page 6, line 225: I would suggest: 'parametric PTF using MNLR' Page 6, line 226: I would suggest: 'which has other inputs than texture and bulk density' Page 6, line 229: I would suggest: 'textural grouping' instead of 'textural classification' (see Page 6, line 234) Page 6, lines 231-232: I would suggest: '...used to develop PTFs from basic soil characteristics to estimate water retention for different textural classes' Page 6, line 235: 'FAO' instead of 'FOA' Page 6, line 238: I would suggest: 'textural grouping' instead of 'textural stratification' Page 6, line 239: I would suggest: 'a better prediction at -1500 kPa was provided by point PTF' Page 7, lines 242-243: I would suggest: 'explained by difficulties in linking water retention properties of the soil samples with their particle size distribution as...' Page 7, line 244: I would suggest: 'After textural grouping, MLR and MNLR PTFs developed are...' Page 7, line 247: I would suggest: 'In the MNLR PTFs' instead of 'Into the MNLR' Page 7, line 259: 'fractions' instead of 'fraction' Page 7, line 260: 'observed' instead of 'observe' Page 7, line 268: 'in all texture classes' instead of 'on all texture classes' Page 7, line 269: 'in the validation dataset' instead of 'in the dataset of validation' Page 7, line 275: 'increases' instead of 'increase' Page 7, line 280: 'low sand content' instead of 'small sand content' Page 8, line 283: 'This is the second most influential variable' Page 8, line 285: 'bulk density' instead of 'the bulk density' Page 8, line 289: 'highly related' instead of 'hugely related' Page 8, line 291: Vertisols Page 8, lines 299-300: 'has a major influence' instead of 'is a major influence' Page 8, lines 301-302: I would suggest: 'predicted values very close to the experimental results are obtained' Page 8, line 303: I would suggest: 'depends on the type of regression techniques' Page 8, line 309: 'PTFs' instead of 'PTF' Page 8, line 312: 'with Clay (%)' instead of 'with the Clay (%)' Page 8, line 315: 'than' instead of 't' Page 8, line 319: I would suggest: 'at high and medium soil water potentials' Page 8, lines 321-322: I would suggest: 'textural grouping' Page 9, line 323: I would suggest: 'The lowest values were recorded' Page 9, line 324: 'of' should be deleted Page 9, lines 330-331: I would suggest: 'textural grouping' Page 9, lines 331-332: I would suggest: 'by the poor OM content in the Algeria soil samples' Page 9, lines 331-332, 333-334: Sometimes 'OM', sometimes 'organic matter'. Please be consistent Page 9, line 333: I would suggest: '...water retention. Danalatos et al. (1994) attributed it to...' Page 9, lines 337-338: I would suggest: 'to
predict θs values' instead of 'to predict saturated soil water contents' Page 9, line 348: 'Indeed' should be deleted Page 9, line 354: I would suggest: 'textural grouping' Page 9, line 355: 'classes' instead of 'class' Page 10, lines 359-360: I would suggest: with clay content > 60% Page 10, line 361: I would suggest: 'textural grouping'

Tables and figures Tables Table 1: - I would suggest as title: 'Soil characteristics of the development and validation datasets' - I would suggest PSD (particle size distribution) instead of Granulometry - 'CV: coefficient of variation' is reported three times on the same table' Table 2: - the line separating MLR and MNLR PTFs is not at the right place - 'Point PTFs' instead of 'Points PTF' - 'multiple R2' instead of 'R2 multiple' - a, b, c,...j are not clearly explained. It would be good to write a general equation with a, b, c,...j as coefficients for more clarity - $\alpha$ should be in kPa-1 - (respectively) should be deleted Table 3: - I would suggest as title: 'Evaluation criteria of water retention pedotransfer functions at -33 kPa and -1500 kPa' Table 4 is missing Table 5: - I would suggest as title: 'Variation of first order sensitivity index along different textural classes' - What does 'Abs' mean? Table 6: - I would suggest as title: 'Pearson correlation matrix between basic soil characteristics in the validation dataset of 53 soil samples'

Figures Each figure caption should be located beneath the respective figure Figures 1, 2, 4, 5 and 6: ',' should be replaced by '.' Figure 2: - I would suggest as title: 'Particle size distribution of xx soil samples from Algeria according to the FAO textural triangle' Figure 3: - Figure 3 is not mentioned in the text Figure 4: - I would suggest as title: 'Root mean square error (RMSE) values calculated for different textural classes' Figure 5: - I would suggest as title: 'Variation of first sensitivity index with RMSE after textural grouping' Figure 6: - Police sizes on the 2 graphs are not the same - 'Point PTF' instead of 'PTF point' - 'Parametric PTF' instead of 'PTF parametric' - 'point' instead of 'ponit'

---

## Author Comment (AC1) · 3 Sep 2016

**First of all, we would like to thank the reviewer for his help improving the paper.**

**Reviewer' comments are in italic, our answers are in bold.**

**S. Barontini (Referee)**

stefano.barontini@unibs.it

1. **My major concerns are about the parametric approach to PTFs.**

*In lines 106 to 113 it is said that the RETC code was used to fit the van Genuchten SWRC to the experimental data, but at line 102 it is reported that the experimental data were determined at two tensiometer–pressure potentials, that are -33 and -1500 kPa. Therefore it seems that the four parameters of the SWRCs ($\theta s$, $\theta r$, $\alpha$, n, while m is constrained to m = 1- 1/n) are fitted by means of two experimetal points only for, each curve.*
*If this is the case, and no other constraints were introduced, the set of parameters is not univocally identified for each soil, and the further analyses on the parametric approach lose their significance.*
*I therefore recommend that either (1) the Authors better detail the followed procedure for this approach, so that it is clear how main experimental points the procedure is based on or whether there were other constraints to univocally identify the fitted parameter set; or (2) they remove the part about the parametric approach and better develop that about the point approach.:* **The text in the lines 106 to 113 was not clear, actually we used the Rosetta model including in RETC code. This sentence will be modified as follows: The van Genuchten's parameters are indirectly estimated for each soil sample from four levels of measured data inputs as: sand, silt and clay percentages and bulk density using H3 Rosetta model (Schaap et al., 2001). The "m" parameter was calculated as follows: m = 1 -1 / n.**
*Moreover (3) I encourage the Authors to explicitly present the PTFs they obtained for the investigated sample of soils. In the following lines, detailed comments and some technical notes will be provided:* **The revised paper will explicitly present the PTF we developed.**

2. **Detailed comments and technical notes**

*ll.37—38 explain whether it refers to the hydrological state of the soil or to the characterization of the hydrological properties:* **It will be Removed**
*l.47 _ ! _ ( (in all the paper):* **It will be modified.**
*l.51 Uniform all the paper to the version "van Genuchten" (or to "Van Genuchten"):* **It will be modified.**
*l.59 "different environments from which they were derived for":* **It will be modified.**
*l.63 "and hydraulic conductivity as well":* **It will be modified.**
*ll.93—96 Check the percentages, or probably better explain the consistency of the whole database:* **It will be modified.**
*l.103 "moisture" -> "water content". Field capacity or soil saturation? Samples in Richards apparatus are usually saturated. Moreover field capacity (regarded to as the soil water content which remains in the soil after abundand imbibition and when percolation is materially decreased) can be quite a small water content, even smaller than the water content at 33 kPa:* **Your comments are right. This will be addressed with text modifications.**
*l.106 "defended"->"defined":* **It will be modified.**
*l.119 Add something like "the following measures of the errors", or something else, to*

*make the article more readable:* **As suggested by the reviewer it will be addressed as: "To discuss the validity of PTF developed, we used the following measures of the errors: the mean prediction error (ME) to inform the bias of the estimate; the mean square error (RMSE)".**

*l.131 Check equation (4), I think that there should not be 1/ n:*
**The index of agreement (Willmott and Wicks, 1980; Willmott, 1981):**

$$d = 1 - \frac{\frac{1}{n}\sum_{i=1}^{n}(\theta_p - \theta_m)^2}{\sum_{i=1}^{n}\left[\left|(\theta_p - \overline{\theta}_m)\right| + \left|(\theta_m - \overline{\theta}_m)\right|\right]^2}$$

**The index of agreement varies from 0 to 1 with higher index values indicating that the modeled values $\theta_p$ have better agreement with the observations $\theta_m$.**

*l.145 Title not necessary:* **Indeed. It will be removed.**

*l.161 Explicit what does the constraint Xi? stand for:* **It will be modified as follows:**
**In order to quantify the variation of sensitivity index (VSi), of an input factor Xi, we can fix it at its"true" value, Xi = Xi ∗ (Xi ∗:the average when the variable follows the normal distribution, the median when the variable follows the lognormal distribution). To calculate how much this assumption change the variance of Y we propose:**
$$V_{Si} = \left(\frac{V[E(Y/X)]}{V(Y)} - \frac{V[E(Y/\,Xi=Xi*)]}{V(Y)}\right) * 100$$
**$V_{Si}$ > 0 and Si close to 1 indicate increasing accuracy of PTFs**
**$V_{Si}$ < 0 and Si close to 1 indicate increasing accuracy of PTFs**
**$V_{Si}$ > 0 and Si close to 0 indicate decreasing accuracy of PTFs**
**$V_{Si}$ < 0 and Si close to 0 indicate decreasing accuracy of PTFs.**

*l.177 Table 2 is not cited before Table 3. This is a good point to explicitly provide the formulae of the obtained PTFs:* **As suggested by the reviewer, It will be modified as follows: We chose to use the Rosetta PTFs in this study because it is one of the latest PTFs and gave reasonable predictions in several evaluation studies (Nemes et al., 2003). The quality prediction of point and parametric PTF developed in this study will be compared with the three Rosetta PTFs (H1, H2, and H3). Three Rosetta models (H1, H2, and H3) were selected because they require the texture and bulk density as inputs as well as the PTF developed (Table 2).**

*ll.199—203 I agree with this sentence, but in this case it can also be due to the undetermination of the interpolated parameters (see the General Comments):* **The discussion will be improved to take this suggestion into account.**

*l.233 Avoid referring to the conductivity as the framework of the article seems to be based on Mualem's predictive approach to the relative conductivity function (as it follows from the constraint on m):* **It will be removed**

*l.244 and followings Consider the idea of collecting all the analyses regarding the texture in one paragraph only, thus restructuring the paragraphs regarding sand, sil and clay. This can strongly help the readability of the discussion. Many analyses of previous Authors are reported: I suggest to explicitly detail whether your results are according or discording to previous ones:* **the reviewer is right. This will be addressed in the revised manuscript.**

*l.291 "They increase in organic matter" with. . . ?.* **It will be removed.**

*l.306 and followings Typically clay is very important at characterising the water retention, even if it can lose sensitivity for great values of clay content: in which sense does it sound the statement of line 317?:* **Indeed, this part will be removed.**

*l.353 I agree with the conclusion but it seems to be quite in contrast to what observed after the reported analyses and the last conclusion: I suggest to better detail this point or remove it.* **As suggested by the reviewer, this part will be removed.**

*Further minor comments: (i) correct some typos, (ii) check the consistency of the references list and alphabetically order it, (iii) change the colour of histograms and bar– graphs to ensure the readibility also in B&W printing:* **All these comments will be addressed in the revised manuscript.**

---

## Author Comment (AC2) · 3 Sep 2016

**The author warmly thanks the reviewer for his/her detailed review. It will considerably improve the manuscript quality. Reviewer' comments are in italic, our answers are in bold.**

Anonymous Referee #2

**Overall opinion**

*This is an interesting paper covering an important topic, namely the prediction of soil hydraulic properties, particularly the soil water retention curve for soils in Algeria. However, there are a number of issues that need to be addressed before publication could be recommended.*

1. **General comments**

*The real major issue that I have with this paper is its lack of novelty. A large number of papers on pedotransfer functions are being submitted to various peer-reviewed journals, which basically all follow the same pattern as this paper does:*
*1. Data are collected locally 2. 'Foreign' PTFs are tested 3. 'Home' PTFs are often developed, but not always 4. Home PTFs are deemed better - or a foreign PTF is found better than others.*
*I agree that using global sensitivity analysis is very useful in decomposing the variance of the response (soil water retention) into contributions from the individual input variables. However, this analysis does not add any new information on what is already known in literature about the contribution of various predictors to the predictive quality of point and parameter-based PTFs. This issue is long known and has been shown/commented on by many papers by now.*
*Another weakness of the paper is its lack of clarity in many parts of the text. I expanded more on this in the specific and technical comments and the authors need to work on that. Good proofreading and editing would considerably improve the quality of the manuscript.* **We also agree with the reviewer that a large number of papers on pedotransfer functions basically discuss the same point about limits on estimates of specific model to a local region or a particular bioclimatic environment when applied in a different context. That's why we developed local pedotransfer functions in this paper; no studies have been conducted before on this subject in Algeria.**
**But beside this quite usual approach; the sensitivity analysis aims at more original objective. The idea is to rank by order of importance the input variables of Algerian FPT such as bulk density, soil texture, and organic matter content at -33 and -1500 kPa in order firstly to detect the contribution of each variable in improving estimates of soil water retention and secondly to identify the source of error and weakness in FPT in each textural class.**
**As a new analytical tool, we proposed to quantify the variation of the first sensitivity index (Vis) after fixing each variable in a central value $Xi = Xi*$ ($Xi*$:the average when the variable follows the normal distribution, the median when the variable follows the lognormal distribution). The variation (Vis) calculated by the corrected formula ( answer: Page 1, line 27) specified in the section (II.2.3) in each textural class for both pressure point gives us an idea about the role of each input in improving the estimate (performance) or in the weakness of FPT (error).**

**What's more we characterized the water retention of Algerians soil by confronting the results of the global sensitivity analysis with research results that address the relationship among the soil water retention and the variables commonly used as input in PTF (Clay, Loam, Sand, bulk density and organic matter).**

**In the revised paper we will include an analysis a part of development of PTF in the section II.**

**2. Specific comments**

*Title: I would suggest: 'Sensitivity analysis of point and parametric pedotransfer functions for estimating water retention of soils in Algeria':* **We totally agree with your suggestion. The title will be changed in the revised manuscript**

*Page 1, line 27: What did the authors mean by: 'favourable impact'?:* **The part of text where we explain how we can assess the impact (lines 160-165) will be modified following the modifications asked by another reviewer:**

**In order to quantify the variation of sensitivity index ($V_{Si}$), of an input factor Xi, we can fix it at its"true" value, $Xi = Xi *$ (Xi ∗: the average when the variable follows the normal distribution, the median when the variable follows the lognormal distribution). To calculate how much this assumption change the variance of Y we propose:**

$$V_{Si} = \left( \frac{V[E(Y/X)]}{V(Y)} - \frac{V[E(Y/ Xi=Xi*)]}{V(Y)} \right) * 100$$

**$V_{Si} > 0$ and Si close to 1 indicate increasing accuracy of PTFs**
**$V_{Si} < 0$ and Si close to 1 indicate increasing accuracy of PTFs**
**$V_{Si} > 0$ and Si close to 0 indicate decreasing accuracy of PTFs**
**$V_{Si} < 0$ and Si close to 0 indicate decreasing accuracy of PTFs**

**We noted an improvement in the estimation of point and parametric PTF in medium textural class at -33 kPa when we have fixed the bulk density or the clay percentage. It means that C% or BD as input in medium textural class at -33 kPa produces more error. The favorable/adverse impact will be removed in the text.**

*Page 1, lines 39-40: I would suggest: 'hydrologists face the situation where soil hydraulic data such as water retention or hydraulic conductivity are often missing. Therefore, pedotransfer functions (PTFs) are used as an alternative to estimate these properties.'* **It will be done as suggested by the reviewer.**

*Page 1, lines 40-41: I do not agree that reports on the evaluation of PTFs outside the area of development are rare (see general comments above). This is one of the main topics in PTF studies.* **Indeed. It will be modified as follows: the extrapolation of PTFs in different agropedoclimatic context limits their performance (Touil et al., 2016).**

*Page 2, line 54: Water retention points are not part of the widespread input data for PTF:* **Indeed. It will be removed. And the sentence will be modified as follows: Schaap et al. (2001) developed the Rosetta package based on the artificial neural network method (ANN), which implements five hierarchical models to predict these parameters with well-defined limits (the soil texture classes only) and the input data (texture, density, and one or two values of water content at -33 and -1500 kPa).**

Page 2, lines 57-58: I am missing something here; why should we call it an advantage? **It will be modified as follow: 97 % are based on multiple linear and polynomial regression of n[th] order techniques (Botula et al. 2014).**

Page 2, line 61: I would expect more recent references:

**Mirus, B. B.,: Evaluating the importance of characterizing soil structure and horizons in parameterizing a hydrologic process model, Hydrological Processes, 29(21), 4611-4623, 2015.**

**The references were also added in the reference list.**

Page 2, line 62: Could the authors provide some references? **We will add the following reference:**
**Bruand, A., Perez-fernandez, P., duval, O., quetin, P., nicoullaud, B., gaillard, H., raison, L., pessaud, J.F. and prud'homme, L. : Estimation des propriétés de rétention en eau des sols: utilisation de classe de pédotransfert après stratifications texturale et texturo-structurale. Etud. Gest. Sols, 9:105-125, 2002.**
**Pachepsky, Y. A., Rawls, W.J.: Soil structure and pedotransfer function, Eur, J. Soil Sci, (54), 443-452, 2003.**
**The references were also added in the reference list.**
*Page 2, line 63: I would expect: 'Soil-water retention and hydraulic conductivity vary widely and non-linearly with soil water potential':* **It will be done as suggested by the reviewer**
Page 2, line 68: Could the authors also provide more recent references:
**Vereecken, H., Feyen, J., Maes, J. and Darius, P.: Estimating the soil moisture retention characteristic from texture, bulk density, and carbon content, Soil Sci, 148,389-403, 1989.**
**Winfield, K.A., Nimmo, J.R., Izbicki, J.A., and MartinResolving, P.M.: Structural Influences on Water-Retention Properties of Alluvial Deposits, Vadose Zone Journal, (5), 706-719, 2006.**
**The references were also added in the reference list.**
*Page 3, lines 85-86: 'comparing the predictive performance with the Rosetta models': this looks like a third objective:* **Indeed, this sentence will be removed.**
*Page 3, line 88: I am missing a short description of the study area and information on soil types:* **The dataset used in this study consists of several subsets collected from some regions in the northern of Algeria. In the revised paper we will include in section II, a part of pedoclimatic information of the north of Algeria.**

*Page 3, line 93: n has been used to design three different variables: (1) number of soil samples in a subset (Page 3, line 93); (2) shape factor of the water retention function (Page 3, line 111); (3) number of horizons (Page 4, line 125):* **we will address these remarks in the revised manuscript as follows:**

**(1) Page 3, line 93 and (Page 3, line 93): The PTFs are developed by using a database collected from some Algerian regions. Subset 1 containing 70 % of the samples from the coastal plain of Annaba located in the north-eastern part of Algeria (13 samples), the plain of Beni Slimane of Media (42 samples), the Kherba El Abadia plain of Ain defla (54 samples) and samples randomly selected from Lower Cheliff plain in northwestern of Algeria (80 samples), soil series was used as the calibration set. Subset 2 with the remaining 21% from Benziane valley in the south west lower Cheliff plain, soil series was selected to verify the PTFs (Table 1). The depth of the two upper horizons varies from site to site with maximum of 30 cm for surface horizons and upper than 30 cm for subsurface horizons. (2) Page 3, line 111: n shape factor of the water retention function. (3) Page 4, line 125: N: number of horizons**

*Page 3, line 96: What did the authors mean by: 'soil series was used as the calibration set'? See also Page 3, line 97, 98:* **It an error, the first sentence (Page 3, line 96) will be removed. The second sentence (Page 3, line 97,98) will be : Subset 2 with the remaining 21% (n = 53) from Benziane valley in the south west lower Cheliff plain, soil series was selected to verify the PTFs 97 (Table 1).**
Page 3, lines 100-105: The authors should provide references for all the lab methods:
**The particle size analysis, conducted using the international Robinson's pipette method (Robinson, 1922). Soil samples taken by cylinders of 500-1000 cm3 (According to the**

case) were used to determine soil bulk density (BD). The water retention values at -33 kPa and -1500 kPa were obtained by Richards's apparatus (Richards, et al., 1943) for samples were collected in moisture nearby to field capacity, by cylinders with a volume of 100 cm3. The water content measurements were conducted by gravimetric method at 105 C° (24h). The organic carbon content was determined by wet oxidation method (Walkley and Black, 1934).
**The references were also added in the reference list.**

*Page 3, lines 102-103: I would rephrase it as follows: 'The water retention values at -33 kPa and -1500 kPa were obtained by the Richards's apparatus. Undisturbed soil samples were collected near field capacity with 100 cm3-cylinders':* **It will be done as suggested by the reviewer.**

*Page 3, line 122: What did the authors mean by:* 'standardised module'?: **It will be modified as follows: the index of agreement (d) developed by Willmott and Wicks (1980), and Willmott (1981) as a standardized measure of the degree of model prediction error.**

*Page 4, line 126: What did the authors mean by: 'The estimate is even less skewed than ME and is close to 0'?:* **It will be modified as follows: with N, number of horizons, θp, θm, predicted and measured volumetric water content, respectively. The estimate is better when ME is close to 0'. Also, negative ME values indicate an average underestimation of θm, while positive values indicate overestimation.**

*Page 4, lines 141-142: ': may manage the functions and non-linear and nonmonotonic models': this sentence is not clear to me. Please rephrase:* **It will be: The Sobol method (Sobol, 1990) is an independent global sensitivity analysis (SA) that is based on decomposition of the variance. When the model is non-linear and non-monotonic, the decomposition of the output variance is still defined and can be used.**

Page 5, line 172: The first reason mentioned by the authors for selecting the Rosetta PTFs in their study seems weak to me as these PTFs have been published 15 years ago. It will be modified as follows: **The prediction quality of point and Parametric PTF developed from Algerian soils are then being compared with the three Rosetta PTFs (H1, H2, and H3). We choose in this work the Rosetta model firstly for the reason of it allows flexibility for the user to input data required (Stumpp et al., 2009) with option of five levels (H1, H2, H3, H4, H5), secondly it is one of the PTF gave reasonable predictions in several evaluation studies (Frederick et al. (2004), Nemes et al., 2003).**
It is of great practical use,

*Page 5, line 175: The authors should give more details on Rosetta models H1, H2 and H3:*
**Three Five hierarchical Rosetta FPT (Schaap et al. 2002) are distinct as five levels based on the input data:**
**H1: The textural classes (USDA classification) ;**
**H2 : Clay+Silt+Sand;**
**H3: Clay+Silt+Sand+ Bulk density;**
**H4: Clay+Silt+Sand+ Bulk density+Volumic water at -33 kPa;**
**H5: Clay+Silt+Sand+ Bulk density+Volumic water at -33 kPa+ Volumic water at -1500 kPa;**

*Page 5, lines 175-176: The second reason for selecting the Rosetta PTFs should be better explained:* **It will be modified as following: The three Rosetta models (H1, H2, and H3) were selected to compare their performance in the Algerian soils because they require only the texture data and bulk density as inputs as well as the locally-developed PTFs**

*Page 5, line 188: What did the authors mean by 'adapt better'?:* **It will be changed by : "give better estimation than"**

*Page 5, line 193: 'Other evaluation criteria noted that the index of agreement also shows that the point PTF is': this sentence is not clear and should be rephrased:* **Furthermore, the**

results index of agreement shows that the point PTF is more suitable for Lower Cheliff soils than the parametric PTF (Fig. 6) with values of (d) (0.9975, 0.9911 cm3 cm -3).

*Page 6, lines 199-200: Did the authors perform a significance test to confirm this?:* **It will be modified as follows: In table 3, no significant difference in RMSE values was observed between the parametric PTFs and Rosetta-H2 at -1500 kPa ( RMSE : 0.0605 cm3 cm-3 and 0.0636 cm3 cm-3 for the parametric PTFs and Rosetta-H2, respectively).**

Page 6, lines 216-217: ': with Si in order to (OM: 0.821; 0.630) and (C %: 0.782; 0.585) at -33 kPa and -1500 kPa, respectively (Fig. 2)': this sentence should be rephrased: **This sentence will be modified as follows: It is clear for the PTFs developed, the organic matter (OM %) and the clay percentages (C %) are the variables that have the most impact particularly on point PTF (MLR) estimation in two pressure points (Si: 0.821; 0.782 at -33 kPa and 0.630; 0.585 at -1500 kPa for the OM and C % respectively.**

*Page 6, line 227: m is directly linked to n by a simple relation (see Page 3, line 113). Therefore, we should only consider 4 parameters: _s, _r, _ and n:* **Indeed. (Parameters of the Van Genuchten curve: Өr, Өs, α, n).**

*Page 7, line 241: What did the authors mean by 'The stability in estimation of PTF before and after classification'?:* **It mean that we didn't observe the improvement in quality estimation of the parametric and point PTFs in the fine and very fine class. This sentence will be modified as follows: The results show that after the textural classification, the improvement of the quality estimation of PTFs is noted only in medium class (Fig. 4). Indeed, a better prediction was recorded by point PTF (RMSE = 0.027 cm3 cm-3) and parametric PTF (RMSE = 0.038 cm3 cm-3) at -1500 kPa.**

*Page 7, lines 265-266: What did the authors mean by: 'when the variation sensitivity index calculated for sand is the leading'? Please rephrase:* **It will be modified as follows: 'when the variation of the first order sensitivity index ($V_{Si}$), for sand is the most important'.**

*Page 7, line 276: What did the authors mean by 'the majority presence'? Please rephrase:* **It will be removed.**

*Page 8, lines 290-294: These 2 sentences are not clear to me. Please rephrase:* **It may also explain the fact that many soils with high clay content in the database are vertisols in which the bulk density and volumetric water content were lower (Rawls et al., 2003). Indeed, the inclusion of the bulk density as input leads to pore volume information, and that can influence the performance of PTFs when they are applied on the soil with high clay content.**

*Page 8, lines 300-301: Which variable do the authors refer to?:* **It will be modified as follows: With the bulk density and the texture as inputs in point PTF (MLR) the nearest experimental results are obtained.**

*Page 8, line 306: What did the authors mean by: 'favourable sensitivity'? Please rephrase:* **In medium texture increasing accuracy of PTFs is noted after fixing the clay content at -33 kPa.**

*Page 8, line 315: What did the authors mean by: 'advanced water retention'? Please rephrase:* **It will be modified as follows: The water retention is higher in very fine and fine classes than the medium class, because it quickly drains water initially retained.**

*Page 8, line 320 to Page 9, line 321: This sentence is not clear to me. How the global sensitivity class and the silt can be estimates of water content? Please rephrase:* **It will be modified as follows : the global sensitivity analysis show that the silt percentage has the second strongest impact on estimation of parametric PTFs at -1500 kPa more than at -33 kPa.**

*Page 9, line 322: What did the authors mean by: 'the main values of $V_{Si}$'? Please rephrase:* **It will be modified as follows: After textural stratification, the important variation of the**

**first order sensitivity index ($V_{Si}$) have been observed in medium class (-36.7% to -1500 kPa).**

*Page 9, line 323: 'or the texture is pure clay': I do not understand this part of the sentence. Please rephrase:* **It will be removed.**

*Page 9, line 325: Do the authors refer to the estimate of water retention or of the van Genucthen parameters?:* **Indeed: It will be modified as follows: It is clear that the percentage of silt has a very important role in estimating the Van Genuchten parameters (α, n), and consequently, its use as input in influences the estimate in the medium and fine class.**

*Page 9, lines 325-327: These 2 sentences are not clear to me. What did the authors mean by 'favourable impact', 'a better pedological interpretation'? Please rephrase:* **It will be modified as follows: Nevertheless, there is an increasing accuracy of PTFs recorded in fine class at -1500 kPa. Its presence with the clay content as inputs has led to a better estimation.**

Page 9, lines 335-336: ': : :as the latter is always considered as the best predictor of soil water retention particularly in clayey soils'. Could the authors support this affirmation by references to recent literature?: **It will be removed.**

*Page 9, line 336: What did the authors mean by 'positive sensitivity impact'? Please rephrase: the* **positive impact mean that the increasing accuracy of PTFs was observed when we had fixed the OM. This sentence will be modified as follows: However, the increasing accuracy of parametric PTFs is observed in medium-textured soils at -33 kPa where the OM is used as input to predict saturated soil water contents.**

Page 9, lines 345-347: This sentence is very weak and does not add on what is already well known from previous studies: **It will be modified as follows:**

**The objective of this study was to analyze the sensitivity of estimating water retention properties of Algerian soil by pedotransfer functions. We presented the development and validation of point and parametric PTFs for estimation of soil hydraulic parameters from basic soil properties regression methods and comparison of the predictive capabilities of these methods with the Rosetta model using some evaluation criteria.**

*Page 9, line 348: ': : :predicts more accuracy than: : :': Please rephrase:* **it will be modified as follows: The reliability tests show that the point PTF produces more accurate estimations than the parametric models.**

*Page 9, lines Tables and figures - Tables and figures should be self-explanatory in their titles and contents. The authors should provide all the necessary information such as explanation for abbreviations, measurement units, etc. Please see also specific comments.*
*- The order of captions of tables and figures should generally correspond with the order*
*of appearance in the text:* **All these comments will be addressed in the revised manuscript**
*- What do the authors mean by cubic model in Table 2?* **It's one of the no linear multiple regression models which we have used to develop the parametric PTFs to estimate the VG parameters. The revised paper will explicitly present the PTF we developed.**

**3. Technical comments**

*Title: I would suggest: Sensitivity analysis of point and parametric pedotransfer functions for estimating water retention of soils in Algeria.* **We agree with the reviewer.**

*Page 1, lines 20-21: the way values of RMSE are reported in the abstract is confusing. Page 1, line 21: RMSE values are in cm3 cm-3 as units for water retention. Page 1, line 28: medium textural class Page 2, lines 44-45: 'estimated' instead of 'constructed' Page 2, line*

*47: the parameters of the van Genuchten model (_s, _r, _ and n) have been introduced before the van Genuchten model itself (line 51). It would be more logical to remove them in line 47 and insert them in line 53.  Page 2, line 53: I would suggest: ': to predict the van Genuchten parameters (_s, _r, _ and n) with soil texture classes only: : :' Page 2, line 54: 'bulk density' instead of 'density' Page 2, line 55: 'Pedotransfer functions' instead of 'PTFs' Page 2, line 60: 'water retention' instead of 'the water retention' Page 2, line 72: 'pedotransfer' should be deleted Page 2, line 76: 'complementary' instead of 'complimentary' Page 3, lines 82-83: I would suggest: 'Deriving and validating two approaches of PTFs using regression methods:' Page 3, line 87: 'input perturbation' is not appropriate. Please rephrase. Page 3, line 92: I would suggest: 'The PTFs are developed using a database of soil samples collected from some regions in Algeria' Page 3, line 93: 'contains' instead of 'containing' Page 3, line 99: 'more than 30 cm' instead of 'upper than 30 cm' Page 3, line 100: 'was conducted using: : :' Page 3, line 100: I would suggest: 'Undisturbed soil samples taken: : :' Page 3, line 101: '(According to the case)' should be deleted Page 3, line 104: I would sug- gest: 'Water content measurements were conducted by the gravimetric method' Page 3, line 106: The word 'defended' is inappropriate. Please rephrase Page 3, lines 111- 112: 'were calculated' instead of 'will be calculated' Page 3, line 117: 'by comparing the values that they predicted' Page 3, lines 118-119: I would suggest: 'To discuss the validity of the PTF developed, we used the following criteria:' Page 3, lines 119-120: 'the root mean square error (RMSE)' instead of 'the mean square error (RMSE)' Page 3, line 120: 'of the quality of the prediction' instead of 'of quality prediction' Page 4, line 129: 'the root mean square error (RMSE)' instead of 'the mean square error (RMSE)' Page 4, lines 143-144: Is it X1 or X1? This needs to be uniform. Page 4, line 144: I would suggest: ': : :X=(X1,: : :, Xp) is the input variable set' Page 4, lines 147-149: The elements of equations 6 and 7 are not explained Page 4, line 157: 'between 0 and 1' instead of 'between [0.1]' Page 5, line 161: What is Xi\* in equation 9? Page 5, line 168: I am missing sentences that introduce Table 1 and Table 2. Moreover, Table 2 is not mentioned in the whole text. Page 5, line 170: I would suggest as title: 'Development of PTFs' Page 5, lines 172-173: With respect to the title in line 170, this sentence should be placed at the end of the paragraph Page 5, line 177: 'soil water retention' instead of 'the soil water retentions' Page 5, line 184: I would suggest: 'the point MLR PTFs' instead of 'the PTF points (MLR)' Page 5, line 185: '0.041 and 0.044' instead of '0,041 and 0,044' Page 5, line 188: 'parametric PTFs' instead of 'parametric PTF' Page 5, line 189: 'neural' instead of 'neuron' Page 5, line 189: 0.0613 and 0.0605 cm3 cm-3 Page 6, line 199: 'while' should be deleted Page 6, line 207: 'PTFs' instead of 'pedotransfer functions' Page 6, line 209: 'fundamental to understand the: : :' instead of 'fundamental to understanding the: : :' Page 6, line 211: 'as bulk density' instead of 'as the bulk density' Page 6, lines 214-217: This sentence is too long and should be divided into 2 sentences. Page 6, lines 215-216: 'at two pressure points' instead of 'in two pressure points' Page 6, line 219: 'in third place' instead of 'in third order' Page 6, line 222: 'MLR' instead of 'linear multiple regression' Page 6, line 224: I would suggest: 'point PTF using MLR is mainly based on: : :' Page 6, line 225: I would suggest: 'parametric PTF using MNLR' Page 6, line 226: I would suggest: 'which has other inputs than tex- ture and bulk density' Page 6, line 229: I would suggest: 'textural grouping' instead of 'textural classification' (see Page 6, line 234) Page 6, lines 231-232: I would suggest: ': : :used to develop PTFs from basic soil characteristics to estimate water retention for different textural classes' Page 6, line 235: 'FAO' instead of 'FOA' Page 6, line 238: I would suggest: 'textural grouping' instead of 'textural stratification' Page 6, line 239: I would suggest: 'a better prediction at -1500 kPa was provided by point PTF' Page 7, lines 242-243: I would suggest: 'explained by difficulties in linking water retention properties of the soil samples with their particle size distribution as...' Page 7, line 244: I would suggest: 'After textural grouping, MLR and MNLR PTFs developed*

*are: : :' Page 7, line 247: I would suggest: 'In the MNLR PTFs' instead of 'Into the MNLR' Page 7, line 259: 'fractions' instead of 'fraction' Page 7, line 260: 'observed' instead of 'observe' Page 7, line 268: 'in all texture classes' instead of 'on all texture classes' Page 7, line 269: 'in the validation dataset' instead of 'in the dataset of validation' Page 7, line 275: 'increases' instead of 'increase' Page 7, line 280: 'low sand content' instead of 'small sand content' Page 8, line 283: 'This is the second most influential variable' Page 8, line 285: 'bulk density' instead of 'the bulk density' Page 8, line 289: 'highly related' instead of 'hugely related' Page 8, line 291: Vertisols Page 8, lines 299-300: 'has a major influence' instead of 'is a major influence' Page 8, lines 301-302: I would suggest: 'predicted values very close to the experimental results are obtained' Page 8, line 303: I would suggest: 'depends on the type of regression techniques' Page 8, line 309: 'PTFs' instead of 'PTF' Page 8, line 312: 'with Clay (%)' instead of 'with the Clay (%)' Page 8, line 315: 'than' instead of 't' Page 8, line 319: I would suggest: 'at high and medium soil water potentials' Page 8, lines 321-322: I would suggest: 'textural grouping' Page 9, line 323: I would suggest: 'The lowest values were recorded' Page 9, line 324: 'of' should be deleted Page 9, lines 330-331: I would suggest: 'textural grouping' Page 9, lines 331-332: I would suggest: 'by the poor OM content in the Algeria soil samples' Page 9, lines 331-332, 333-334: Sometimes 'OM', sometimes 'organic matter'. Please be consistent Page 9, line 333: I would suggest: '...water retention. Danalatos et al. (1994) attributed it to...' Page 9, lines 337-338: I would suggest: 'to predict _s values' instead of 'to predict saturated soil water contents' Page 9, line 348: 'Indeed' should be deleted Page 9, line 354: I would suggest: 'textural grouping' Page 9, line 355: 'classes' instead of 'class' Page 10, lines 359-360: I would suggest: with clay content > 60% Page 10, line 361: I would suggest: 'textural grouping' Tables and figures Tables Table 1: - I would suggest as title: 'Soil characteristics of the development and validation datasets' - I would suggest PSD (particle size distribution) instead of Granulometry - 'CV: coefficient of variation' is reported three times on the same table' Table 2: - the line separating MLR and MNLR PTFs is not at the right place - 'Point PTFs' instead of 'Points PTF' - 'multiple R2' instead of 'R2 multiple' - a, b, c,...j are not clearly explained. It would be good to write a general equation with a, b, c,...j as coefficients for more clarity - _ should be in kPa-1 - (respectively) should be deleted Table 3: - I would suggest as title: 'Evaluation criteria of water retention pedotransfer functions at -33 kPa and -1500 kPa' Table 4 is missing Table 5: - I would suggest as title: 'Variation of first order sensitivity index along different textural classes' - What does 'Abs' mean? Table 6: - I would suggest as title: 'Pearson correlation matrix between basic soil characteristics in the validation dataset of 53 soil samples' Figures Each figure caption should be located beneath the respective figure Figures 1, 2, 4, 5 and 6: ',' should be replaced by '.' Figure 2: - I would suggest as title: 'Particle size distribution of xx soil samples from Algeria according to the FAO textural triangle' Figure 3: - Figure 3 is not mentioned in the text Figure 4: - I would suggest as title: 'Root mean square error (RMSE) values calculated for different textural classes' Figure 5: - I would suggest as title: 'Variation of first sensitivity index with RMSE after textural grouping' Figure 6: - Police sizes on the 2 graphs are not the same - 'Point PTF' instead:* **All these comments will be addressed in the revised manuscript.**

**Additional References**

**Aubert, G. : Méthodes d'analyses des sols, Edit.C.R.D.P. Marseille, 189, 1978.**

**Botula, Y. D., Van Ranst, E., Cornelis, W.M.: Pedotransfer functions to predict water retention of soils from the humid tropics: A review, Brazilian Journal of Soil Science, 38, 679-698, 2014.**

BRUAND, A., PEREZ-FERNANDEZ, P., DUVAL, O., QUETIN, P., NICOULLAUD, B., GAILLARD, H., RAISON, L., PESSAUD, J.F. and PRUD'HOMME, L. : Estimation des propriétés de rétention en eau des sols: utilisation de classe de pédotransfert après stratifications texturale et texturo-structurale. Etud. Gest. Sols, 9:105-125, 2002.

Mirus, B. B.,: Evaluating the importance of characterizing soil structure and horizons in parameterizing a hydrologic process model, Hydrological Processes, 29(21), 4611-4623, 2015.

Pachepsky. Y. A., Rawls, W.J.: Soil structure and pedotransfer function,Eur,J,Soil Sci, (54), 443-452,2003

Richards, L. A., and Fireman, M..: Pressure-plate apparatus for measuring moisture sorption and transmission by soils. Soil Sci. 56: 395–404, 1943.

Robinson, G. W.: A new method for the mechanical analysis of soils and other dispersions. J. Agric. Sci., 12, 1922.

Stumpp, C., Engelhardt, S., Hofmann, M., and Huwe, B.: Evaluation of Pedotransfer Functions for Estimating Soil Hydraulic Properties of Prevalent Soils in a Catchment of the Bavarian Alps, European Journal of Forest Research, 128(6), 609-620, 2009.

Vereecken, H., Feyen, J., Maes, J. and Darius, P.: Estimating the soil moisture retention characteristic from texture, bulk density, and carbon content, Soil Sci, 148,389-403, 1989.

Walkley, A. and Black, I.A.: An examination of the Degtjareff method for determining organic carbon in soils: Effect of variations in digestion conditions and of inorganic soil constituent, Soil Sci, 63, 251-263, 1934.

Winfield, K.A., Nimmo, J.R., Izbicki, J.A., and MartinResolving, P.M.: Structural Influences on Water-Retention Properties of Alluvial Deposits, Vadose Zone Journal, (5), 706-719, 2006.

---

## Author Response (AR1)

Dear Dr David Dunkerley (Topical Editor),

We thank the Topical Editor and reviewers for their comments and suggestions that helped to improve the quality of the manuscript.

Please find below the point-by-point response to the reviewers' comments. Reviewer' comments are in italic, the authors' replies are in bold.

Following the suggestion of Referee #2; the manuscript title has been changed to: '**Sensitivity analysis of point and parametric pedotransfer functions for estimating water retention of soils in Algeria'.**

Some of the comments were general across more than one referee (R) and these are treated together (I, II, and III below) in a joint reply to the tow referees:

**I. Database:** all two reviewers have an opinion about this part. *R1: In lines 106 to 113 it is said that the RETC code was used to fit the van Genuchten SWRC to the experimental data, but at line 102 it is reported that the experimental data were determined at two tensiometer–pressure potentials, that are -33 and -1500 kPa. Therefore it seems that the four parameters of the SWRCs (θs, θr, α, n, while m is constrained to m = 1- 1/n) are fitted by means of two experimetal points only for, each curve.*
*If this is the case, and no other constraints were introduced, the set of parameters is not univocally identified for each soil, and the further analyses on the parametric approach lose their significance.*
*I therefore recommend that either (1) the Authors better detail the followed procedure for this approach, so that it is clear how main experimental points the procedure is based on or whether there were other constraints to univocally identify the fitted parameter set; or (2) they remove the part about the parametric approach and better develop that about the point approach.* ***R1**: l.103"moisture" - >"water content". Field capacity or soil saturation? Samples in Richards's apparatus are usually saturated. Moreover field capacity (regarded to as the soil water content which remains in the soil after abundand imbibition and when percolation is materially decreased) can be quite small water content, even smaller than the water content at 33 kPa.* ***R1**: ll.199—203 I agree with this sentence, but in this case it can also be due to the undetermination of the interpolated parameters (see the General Comments).* ***R2**: Page 3, line 88: I am missing a short description of the study area and information on soil types.* ***R2**: Page 3, lines 102-103: I would rephrase it as follows: 'The water retention values at -33 kPa and -1500 kPa were obtained by the Richards's apparatus. Undisturbed soil samples were collected near field capacity with 100 cm3-cylinders'.* ***R2**: Page 3, lines 100-105: The authors should provide references for all the lab methods.*

*Our reply:* **This part will be addressed in the revised paper as follows (see page 1 to 3, lines 85 to 116):**

**The soil dataset used for this study was collected from various regions in Algeria, mainly in the north, which has a Mediterranean climate. It contained 242 samples, with basic soil properties: texture fractions (based on the USDA system; clay and silty-clayey for most of the soils, Fig. 3a), BD, OM content and water content at -33 kPa and -1500kPa. Descriptive statistics of the development and validation datasets are presented in Table 1. The available database was split into two datasets. Subset 1, which was used to develop the PTFs, contained 78.1% of the samples. Used as the calibration set, they were collected from the coastal plain of Annaba in north-eastern Algeria (13 samples), the Beni Slimane plain of Media (42 samples), the Kherba El Abadia plain of Ain Defla (54 samples) and the Lower Cheliff plain in north-western Algeria (80 samples). Subset 2 contained the remaining 21.9% of the samples. Used to verify the PTFs, they were collected from Benziane valley in the lower south-western Cheliff plain. The depth of the two upper horizons varied from site to site, with a maximum of 30 cm for surface horizons and more than 30 cm for subsurface horizons.**

**PSD analysis was conducted using the international Robinson's pipette method (Robinson, 1922). Undisturbed soil samples obtained with 500-1,000 cm3 cylinders were used to determine BD. The SWR values at -33 kPa and -1500 kPa were obtained using Richards's apparatus (Richards et al., 1943). Undisturbed soil samples were collected near field capacity with 100 cm3 cylinders. Water content was measured using the gravimetric method at 105°C (24 h). Organic carbon content was determined using the wet oxidation method (Walkley and**

**Black, 1934). Variation in soil texture in the dataset is displayed using the textural triangle proposed by FAO (1990) in Figure 3b.**
**The SWR model devised by Van Genuchten (1980) is defined as:**

$$\theta(h) = \theta_r + \frac{\theta_s - \theta_r}{(1+|\alpha h|^n)^m} \tag{1}$$

**Where $\theta_r$ and $\theta_s$ are residual and saturated soil-water content (cm3 cm–3), respectively, and $\alpha$ (cm–1) and n are the shape factors of the SWR function. The VG parameters were indirectly estimated for each soil sample from four levels of measured data inputs: sand, silt and clay percentages, and BD using the Rosetta model H3 (Schaap et al., 2001). The 'm' parameter was calculated as follows: m = 1 -1 / n.**

**The references were also added in the reference list.**

**II. Development of PTFs:** ***R1****: I encourage the Authors to explicitly present the PTFs they obtained for the investigated sample of soils.****R1****: l.177 Table 2 is not cited before Table 3. This is a good point to explicitly provide the formulae of the obtained PTFs. ****R2****: lines 147-149: The elements of equations 6 and 7 are not explained Page 4. ****R2****: What do the authors mean by cubic model in Table 2?.*

**Our reply:** **This part will be addressed in the revised paper as follows (see page 4 lines 117 to 145):**
**Two approaches were used in this study to develop the PTFs: point PTFs for estimating SWR for particular points of pressure (h); and parametric PTFs for predicting the VG parameters. Each water content level at selected water potentials of -33 kPa and -1500 kPa and estimated VG parameters were related to basic soil properties (i.e., sand, silt, clay content, OM content and BD) using multiple regression techniques (Table 2). The most significant input variables were determined using the Pearson correlation ($\alpha$ =5%). T For the multiple-linear regression (MLR) models, the general form of the resulting equations was expressed thus:**
**$Y = a_0 + b_1 X_1 + b_2 X_2 + b_3 X_3 + b_4 X_4$** (2)
**For the multiple-non-linear regression (MNLR) models, it was expressed thus:**
**$Y = a_0 + b_1 X_1 + b_2 X_2 + b_3 X_1{}^2 + b_4 X_2{}^2 + b_5 X_1{}^3 + b_6 X_2{}^3 + b_7 X_1{}^* X_2 + b_8 X_1{}^2{}^* X_2 + b_9 X_1{}^* X_2{}^2$**
**(3)**
**Where Y represents the dependent variable, $a_0$ is the intercept; $b_1$..., bn are the regression coefficients, and X1 to X4 refer to the independent variables representing the basic soil properties.**

**The prediction quality of the point and parametric PTFs developed from Algerian soils were then compared with three Rosetta PTFs (H1, H2 and H3). We chose the Rosetta model because it gives the user flexibility in inputting the data required (Stumpp et al., 2009), with the option of five levels based on input data (Schaap et al. 2002):**
- **H1: Textural classes (USDA system);**
- **H2 : Clay+Silt+Sand;**
- **H3: Clay+Silt+Sand+ BD;**
- **H4: Clay+Silt+Sand+ BD +Volumic water at -33 kPa;**
- **H5: Clay+Silt+Sand+ BD +Volumic water at -33 kPa + Volumic water at -1500 kPa.**
**The Rosetta model was also chosen because it has given reasonable predictions in several evaluation studies (Frederick et al., 2004, Nemes et al., 2003). In our study, the three Rosetta model levels (H1, H2, and H3) were selected to compare their performance in the Algerian soils because they require only texture data and BD as inputs, as well as the locally developed PTFs.**

**III. Global sensitivity analysis (GSA) approach:** ***R1****: l.161 Explicit what does the constraint Xi? stand for. ****R1****: l.145 Title not necessary. ****R2****: Page 1, line 27: What did the authors mean by: 'favourable impact'?. ****R2****: Page 4, lines 141-142: ': may manage the functions and non-linear and nonmonotonic models': this sentence is not clear to me. Please rephrase.*

**Our reply:** **This part will be addressed in the revised paper as follows (see page 1 to 3, lines 165 to 204):**

GSA involves determining which part of the variance in model response is due to variance in which input variable or group of inputs. The impact of the parameters is quantified by calculating the global sensitivity indices.

The Sobol method (Sobol, 1990) is an independent GSA method based on decomposition of the variance. When the model is non-linear and non-monotonic, the decomposition of the output variance is still defined and can be used. The Sobol model is represented by the following function:

$$Y= f (X_1, X_2, X_3,........,X_p) \tag{7}$$

Where Y is the model output (or objective function) and $X=(X_1,....., X_p)$ is the input variable set.

$$V(Y) = V (E (Y|X)) + E (Var (Y|X)) \tag{8}$$

Where V(Y) is the total variance in the model, V (E(Y|X)) and E (Var(Y|X)) signify variance in the conditional expected value and expected value of the conditional variance, respectively. When the input variables $X_i$ are independent, the variance decomposition of the model is:

$$V(Y) = \sum_{i=1}^{p} V_i + \sum_i \sum_j V_{ij} + + \sum_i \sum_j \sum_p V_{ijp} + \ldots\ldots\ldots + V_{1,2,3,\ldots p}$$

(9)

$$V_i = V [ E(Y|X_i)]$$

$$V_{ij} = V [ E(Y|X_i, X_j)] -V_i-V_j$$

$$V_{ijp}= V [ E(Y|X_i, X_j, X_p)] - V_{ij} - V_{ip} - V_{jp} - V_i - V_j - V_p$$

Where $V_i$ is the proportion of variance due to variable $X_i$. Dividing $V_i$ by V(Y) produces the expression of the first-order sensitivity index ($S_i$), such that:

$$S_i = \frac{V_i}{V(Y)} = \frac{V[E(Y/X_i)]}{V(Y)} \tag{10}$$

The term $S_i$ is the measure that guarantees an informed choice in cases where the factors are correlated and interact (Saltelli and Tarantola, 2002). This index is always between 0 and 1, and represents a proper measurement of the sensitivity used to classify the input variables in order of importance (Saltelli and Tarantola, 2001).

In order to quantify variation in the sensitivity index ($V_{Si}$) of an input factor $X_i$, we fixed it at $Xi = Xi*$ ($Xi*$: the average when the variable follows the normal distribution, the median when the variable follows the lognormal distribution). In order to calculate how much this assumption changed the variance of Y, we used this formula:

$$V_{Si} = \left( \frac{V[E(Y/X)]}{V(Y)} - \frac{V[E(Y/ Xi=Xi*)]}{V(Y)} \right) * 100 \tag{11}$$

$V_{Si} > 0$ and $S_i$ close to 1 indicate increasing accuracy of PTFs;

$V_{Si} < 0$ and $S_i$ close to 1 indicate increasing accuracy of PTFs;

$V_{Si} > 0$ and $S_i$ close to 0 indicate decreasing accuracy of PTFs;

$V_{Si} < 0$ and $S_i$ close to 0 indicate decreasing accuracy of PTFs.

In addition, combining the RMSE and $S_i$ enabled us to detect the contribution of each variable to improvement in the quality of prediction of the PTFs.

**Some additional referee comments were raised**

**Referee # 1 (Dr. S. Barontini**)

    1. **Detailed comments and technical notes**

*ll.37—38 explain whether it refers to the hydrological state of the soil or to the characterization of the hydrological properties:* **It will be removed**
*l.47 _ ! _ ( (in all the paper):* **It will be modified.**
l.*51 Uniform all the paper to the version "van Genuchten" (or to "Van Genuchten"):* **It will be modified.**

*l.59 "different environments from which they were derived for":* **It will be modified.**
*l.63 "and hydraulic conductivity as well":* **It will be modified.**
*l.106 "defended"->"defined":* **It will be modified.**
*l.119 Add something like "the following measures of the errors", or something else, to*
*make the article more readable:* **As suggested by the reviewer it will be addressed as: "In order to assess the validity of the PTFs developed, we used the following criteria: mean prediction error (ME) to indicate the bias of the estimate; root mean square error (RMSE) to assess the quality of the prediction (it is frequently used in studies on PTFs); and the index of agreement (d) developed by Willmott and Wicks (1980) and Willmott (1981) as a standardized measure of the degree of model prediction error.(see page 4 lines 147 to 152).**
*l.131 Check equation (4), I think that there should not be 1/ n:*
**The index of agreement (Willmott and Wicks, 1980; Willmott, 1981):**

$$d = 1 - \frac{\frac{1}{n}\sum_{i=1}^{n}(\theta_{\mathrm{p}} - \theta_{\mathrm{m}})^2}{\sum_{i=1}^{n}\left[\left|(\theta_{\mathrm{p}} - \overline{\theta}_{\mathrm{m}})\right| + \left|(\theta_{\mathrm{m}} - \overline{\theta}_{\mathrm{m}})\right|\right]^2}$$

**The index of agreement varies from 0 to 1 with higher index values indicating that the modeled values $\theta_{\mathrm{p}}$ have better agreement with the observations $\theta_{\mathrm{m}}$. (See page 4 to 3 lines 159 to 163).**
*l.145 Title not necessary:* **Indeed. It will be removed.**
*l.233 Avoid referring to the conductivity as the framework of the article seems to be based on Mualem's predictive approach to the relative conductivity function (as it follows from the constraint on m):* **It will be removed**
*l.244 and followings Consider the idea of collecting all the analyses regarding the texture in one paragraph only, thus restructuring the paragraphs regarding sand, sil and clay. This can strongly help the readability of the discussion. Many analyses of previous Authors are reported: I suggest to explicitly detail whether your results are according or discording to previous ones:* **the reviewer is right. This will be addressed in the revised manuscript (see page 7 to 9 lines 260 to 345).**
*l.291 "They increase in organic matter" with. . . ?.* **It will be removed.**
*l.306 and followings Typically clay is very important at characterising the water retention, even if it can lose sensitivity for great values of clay content: in which sense does it sound the statement of line 317?:* **Indeed, this part will be removed.**
*l.353 I agree with the conclusion but it seems to be quite in contrast to what observed after the reported analyses and the last conclusion: I suggest to better detail this point or remove it.* **As suggested by the reviewer, this part will be removed.**
*Further minor comments: (i) correct some typos, (ii) check the consistency of the references list and alphabetically order it, (iii) change the colour of histograms and bar– graphs to ensure the readibility also in B&W printing:* **All these comments will be addressed in the revised manuscript.**

**Anonymous Referee #2**

**Overall opinion**

*This is an interesting paper covering an important topic, namely the prediction of soil hydraulic properties, particularly the soil water retention curve for soils in Algeria. However, there are a number of issues that need to be addressed before publication could be recommended.*

**1. General comments**

*The real major issue that I have with this paper is its lack of novelty. A large number of papers on pedotransfer functions are being submitted to various peer-reviewed journals, which basically all follow the same pattern as this paper does:*
*1. Data are collected locally 2. 'Foreign' PTFs are tested 3. 'Home' PTFs are often developed, but not always 4. Home PTFs are deemed better - or a foreign PTF is found better than others.*
*I agree that using global sensitivity analysis is very useful in decomposing the variance of the response (soil water retention) into contributions from the individual input variables. However, this analysis does not add any new information on what is already known in literature about the contribution of various predictors to the predictive quality of point and parameter-based PTFs. This issue is long known and has been shown/commented on by many papers by now.*

*Another weakness of the paper is its lack of clarity in many parts of the text. I expanded more on this in the specific and technical comments and the authors need to work on that. Good proofreading and editing would considerably improve the quality of the manuscript.*

*Our reply:* **We also agree with the reviewer that a large number of papers on pedotransfer functions basically discuss the same point about limits on estimates of specific model to a local region or a particular bioclimatic environment when applied in a different context. That's why we developed local pedotransfer functions in this paper; no studies have been conducted before on this subject in Algeria.**

**But beside this quite usual approach; the sensitivity analysis aims at more original objective. The idea is to rank by order of importance the input variables of Algerian FPT such as bulk density, soil texture, and organic matter content at -33 and -1500 kPa in order firstly to detect the contribution of each variable in improving estimates of soil water retention and secondly to identify the source of error and weakness in FPT in each textural class.**

**As a new analytical tool, we proposed to quantify the variation of the first sensitivity index (Vis) after fixing each variable in a central value $Xi = Xi*$ ($Xi*$:the average when the variable follows the normal distribution, the median when the variable follows the lognormal distribution). The variation (Vis) calculated by the corrected formula ( answer: Page 1, line 27) specified in the section (II.2.3) in each textural class for both pressure point gives us an idea about the role of each input in improving the estimate (performance) or in the weakness of FPT (error).**

**What's more we characterized the water retention of Algerians soil by confronting the results of the global sensitivity analysis with research results that address the relationship among the soil water retention and the variables commonly used as input in PTF (Clay, Loam, Sand, bulk density and organic matter).**

**In the revised paper we will include an analysis a part of development of PTF in the section II.**

**2. Specific comments**

*Page 1, lines 39-40: I would suggest: 'hydrologists face the situation where soil hydraulic data such as water retention or hydraulic conductivity are often missing. Therefore, pedotransfer functions (PTFs) are used as an alternative to estimate these properties.'* **It will be done as suggested by the reviewer (see page 1, lines 37-39).**

*Page 1, lines 40-41: I do not agree that reports on the evaluation of PTFs outside the area of development are rare (see general comments above). This is one of the main topics in PTF studies.* **Indeed. It will be modified as follows: the extrapolation of PTFs in different agropedoclimatic context limits their performance (Touil et al., 2016). (See page 1, lines 39-40).**

*Page 2, line 54: Water retention points are not part of the widespread input data for PTF:* **Indeed. It will be removed. And the sentence will be modified as follows: Schaap et al. (2001) developed the Rosetta package based on the artificial neural network method (ANN), which implements five hierarchical models to predict these parameters with well-defined limits (the soil texture classes only) and the input data (texture, density, and one or two values of water content at -33 and -1500 kPa).**

*Page 2, lines 57-58: I am missing something here; why should we call it an advantage?* **It will be modified as follow: 97 % are based on multiple linear and polynomial regression of $n^{th}$ order techniques (Botula et al. 2014). (See page 2, lines 55-56).**

*Page 2, line 61: I would expect more recent references:*
**Mirus, B. B.,: Evaluating the importance of characterizing soil structure and horizons in parameterizing a hydrologic process model, Hydrological Processes, 29(21), 4611-4623, 2015.**
**The references were also added in the reference list. (See page 2, line 59).**

*Page 2, line 62: Could the authors provide some references?* **We will add the following reference (See page 2, line 61):**
**Bruand, A., Perez-fernandez, P., duval, O., quetin, P., nicoullaud, B., gaillard, H., raison, L., pessaud, J.F. and prud'homme, L. : Estimation des propriétés de rétention en eau des sols: utilisation de classe de pédotransfert après stratifications texturale et texturo-structurale. Etud. Gest. Sols, 9:105-125, 2002.**
**Pachepsky, Y. A., Rawls, W.J.: Soil structure and pedotransfer function, Eur, J. Soil Sci, (54), 443-452, 2003.**
**The references were also added in the reference list.**

*Page 2, line 63: I would expect: 'Soil-water retention and hydraulic conductivity vary widely and non-linearly with soil water potential':* **It will be done as suggested by the reviewer. (See *page 2, line 62).***

Page 2, line 68: Could the authors also provide more recent references: **We will add the following reference (See Page 2, line 67):**
**Vereecken, H., Feyen, J., Maes, J. and Darius, P.: Estimating the soil moisture retention characteristic from texture, bulk density, and carbon content, Soil Sci, 148,389-403, 1989.**
**Winfield, K.A., Nimmo, J.R., Izbicki, J.A., and MartinResolving, P.M.: Structural Influences on Water-Retention Properties of Alluvial Deposits, Vadose Zone Journal, (5), 706-719, 2006.**
**The references were also added in the reference list.**
*Page 3, lines 85-86: 'comparing the predictive performance with the Rosetta models': this looks like a third objective:* **Indeed, this sentence will be removed.**
*Page 3, line 93: n has been used to design three different variables: (1) number of soil samples in a subset (Page 3, line 93); (2) shape factor of the water retention function (Page 3, line 111); (3) number of horizons (Page 4, line 125):* **we will address these remarks in the revised manuscript as follows:**

**(1) Page 3, line 93: Subset 1, which was used to develop the PTFs, contained 78.1% of the samples. Used as the calibration set, they were collected from the coastal plain of Annaba in north-eastern Algeria (13 samples), the Beni Slimane plain of Media (42 samples), the Kherba El Abadia plain of Ain Defla (54 samples) and the Lower Cheliff plain in north-western Algeria (80 samples). Subset 2 contained the remaining 21.9% of the samples. Used to verify the PTFs, they were collected from Benziane valley in the lower south-western Cheliff plain (see page 3, line 95). (2) Page 3, line 111: n shape factor of the water retention function (see page 3, line 114). (3) Page 4, line 125: N: number of horizons (see page 4, line 155).**

*Page 3, line 96: What did the authors mean by: 'soil series was used as the calibration set'? See also Page 3, line 97, 98:* **It an error, the first sentence (Page 3, line 96) will be removed. The second sentence (Page 3, line 97, 98) will be: Subset 2 contained the remaining 21.9% of the samples. Used to verify the PTFs, they were collected from Benziane valley in the lower south-western Cheliff plain (see page 3, lines 98-100).**
*Page 3, line 122: What did the authors mean by:* 'standardised module'?: **It will be modified as follows: the index of agreement (d) developed by Willmott and Wicks (1980), and Willmott (1981) as a standardized measure of the degree of model prediction error (see Page 3, line 151).**
*Page 4, line 126: What did the authors mean by: 'The estimate is even less skewed than ME and is close to 0'?:* **It will be modified as follows: with N, number of horizons, θp, θm, predicted and measured volumetric water content, respectively. The estimate is better when ME is close to 0'. Also, negative ME values indicate an average underestimation of θm, while positive values indicate overestimation (see page 4, line 156-158).**
*Page 5, line 172: The first reason mentioned by the authors for selecting the Rosetta PTFs in their study seems weak to me as these PTFs have been published 15 years ago.* **It will be modified as follows: The prediction quality of the point and parametric PTFs developed from Algerian soils were then compared with three Rosetta PTFs (H1, H2 and H3). We chose the Rosetta model because it gives the user flexibility in inputting the data required (Stumpp et al., 2009), with the option of five levels based on input data (Schaap et al. 2002). (See page 4 lines 133-136).**
*Page 5, line 175: The authors should give more details on Rosetta models H1, H2 and H3:*
**The Rosetta FPTs (Schaap et al. 2002) are distinct as five levels based on the input data (see page 4 lines 136-141):**
**H1: The textural classes (USDA classification):**
**H2: Clay+Silt+Sand**
**H3: Clay+Silt+Sand+ Bulk density**
**H4: Clay+Silt+Sand+ Bulk density+Volumic water at -33 kPa**
**H5: Clay+Silt+Sand+ Bulk density+Volumic water at -33 kPa+ Volumic water at -1500 kPa**
*Page 5, lines 175-176: The second reason for selecting the Rosetta PTFs should be better explained:* **It will be modified as following: The Rosetta model was also chosen because it has given reasonable predictions in several evaluation studies (Frederick et al., 2004, Nemes et al., 2003). In our study, the three Rosetta model levels (H1, H2, and H3) were selected to compare their performance in the Algerian soils because they require only texture data and BD as inputs, as well as the locally developed PTFs (see page 4 lines 143-146).**
*Page 5, line 188: What did the authors mean by 'adapt better'?:* **It will be changed by : "gave a better estimation than" (See page 6, line 219).**
*Page 5, line 193: 'Other evaluation criteria noted that the index of agreement also shows that the point PTF is': this sentence is not clear and should be rephrased:* **It will be modified as follows : The index of agreement results showed that point PTFs were more suitable for Lower Cheliff soils**

than parametric PTFs (Figure 6), with values of 0.9975 and 0.9911 cm$^3$ cm$^{-3}$). (See page 6 lines 223-224).

*Page 6, lines 199-200: Did the authors perform a significance test to confirm this?:* **It will be modified as follows: As Table 3 shows, there was no significant difference in RMSE values between the parametric PTFs and Rosetta H2 at -1500 kPa (RMSE: 0.0605 cm$^3$ cm$^{-3}$ and 0.0636 cm$^3$ cm$^{-3}$, respectively).(See page 6 lines 228-229).**

Page 6, lines 216-217: ': with Si in order to (OM: 0.821; 0.630) and (C %: 0.782; 0.585) at -33 kPa and -1500 kPa, respectively (Fig. 2)': this sentence should be rephrased: **This sentence will be modified as follows: It was clear for the PTFs developed that OM% and clay percentages (C%) were the variables with the greatest impact (Figure 2). For the point PTFs (MLR), the most sensitive estimations were at two pressure points (S$_i$: 0.821; 0.782 at -33 kPa and 0.630; 0.585 at -1500 kPa for OM% and C%, respectively. (See page 7 lines 237-240).**

*Page 6, line 227: m is directly linked to n by a simple relation (see Page 3, line 113). Therefore, we should only consider 4 parameters: Өs, Өr, α and n:* **Indeed. (VG parameters: $\theta_r$, $\theta_s$, α, n). (See page 7 lines 248).**

*Page 7, line 241: What did the authors mean by 'The stability in estimation of PTF before and after classification'?:* **It mean that we didn't observe the improvement in quality estimation of the parametric and point PTFs in the fine and very fine class. This sentence will be modified as follows: The results showed that after the textural grouping, there was an improvement in the quality estimation of PTFs only in the medium class (Figure 4). A better prediction at -1500 kPa was provided by point PTFs (RMSE = 0.027 cm$^3$ cm$^{-3}$) and parametric PTFs (RMSE = 0.038 cm$^3$ cm$^{-3}$) at -1500 kPa. (See page 7 lines 257-259).**

*Page 7, lines 265-266: What did the authors mean by: 'when the variation sensitivity index calculated for sand is the leading'? Please rephrase:* **It will be modified as follows: when the variation of the first order S$_i$ for sand was the most important (see page 8 lines 278).**

*Page 7, line 276: What did the authors mean by 'the majority presence'? Please rephrase:* **It will be removed.**

*Page 8, lines 290-294: These 2 sentences are not clear to me. Please rephrase:* **It will be modified as follows: This might also explain the fact that many soils with high clay content in the database are Vertisols in which BD and VWC are lower (Rawls et al., 2003). The inclusion of BD as an input provides information on pore volume, which can influence the performance of PTFs when applied to soil with high clay content (see page 9 lines page 322-325).**

*Page 8, lines 300-301: Which variable do the authors refer to?:* **It will be modified as follows: With BD and texture as inputs in point PTF (MLR), predicted values very close to the experimental results are obtained (see page 9 lines 231-232).**

*Page 8, line 306: What did the authors mean by: 'favourable sensitivity'? Please rephrase:* **In the medium texture class, there was increasing accuracy in PTFs after fixing the clay content at -33 kPa (see page 8 lines 295-296).**

*Page 8, line 315: What did the authors mean by: 'advanced water retention'? Please rephrase:* I**t will be modified as follows: SWR was higher in the very fine and fine classes than in the medium class, because they quickly drained water initially retained (see page 8 lines 303-305).**

*Page 8, line 320 to Page 9, line 321: This sentence is not clear to me. How the global sensitivity class and the silt can be estimates of water content? Please rephrase:* I**t will be modified as follows : The GSA showed that the silt percentage had a stronger impact on the estimation of parametric PTFs at -1500 kPa than at -33 kPa with the MNLR model (see page 8 lines 307-309).**

*Page 9, line 322: What did the authors mean by: 'the main values of V$_{Si}$'? Please rephrase:* I**t will be modified as follows:  After textural grouping, an important variation in the first order S$_i$ was observed in the medium class (-36.7% to -1500 kPa). (See page 8 lines 309-310).**

*Page 9, line 323: 'or the texture is pure clay': I do not understand this part of the sentence. Please rephrase:* **It will be removed.**

*Page 9, line 325: Do the authors refer to the estimate of water retention or of the van Genucthen parameters?:* **Indeed: It will be modified as follows: It was clear that the silt percentage has an important role in estimating VG's parameters (α, n), and that its use as an input influences the estimate in the medium and fine classes (see page 8, 9 lines 311-312).**

*Page 9, lines 325-327: These 2 sentences are not clear to me. What did the authors mean by 'favourable impact', 'a better pedological interpretation'? Please rephrase:* **It will be modified as follows:  There was an increasing accuracy, however, in the PTFs recorded in the fine class at -1500 kPa. With silt and clay as inputs, there was a better estimation (see page 9 lines 313-314).**

Page 9, lines 335-336: ': : :as the latter is always considered as the best predictor of soil water retention particularly in clayey soils'. Could the authors support this affirmation by references to recent literature?: **It will be removed.**

*Page 9, line 336: What did the authors mean by 'positive sensitivity impact'? Please rephrase:* **This sentence will be modified as follows: the increasing accuracy of parametric PTFs, however, was apparent for medium-textured soils at -33 kPa, where OM was used as an input to predict $\theta_s$ (see page 9 lines 339-341).**

Page 9, lines 345-347: This sentence is very weak and does not add on what is already well known from previous studies: **It will be modified as follows:**

**The objective of this study was to analyze the sensitivity of estimating the SWR properties of Algerian soil using PTFs. We developed and validated point and parametric PTFs from basic soil properties using regression techniques and compared their predictive capabilities with the Rosetta models (H1, H2, and H3). (See page 9-10 lines 348-352).**

*Page 9, line 348: ': : :predicts more accuracy than: : :': Please rephrase:* **it will be modified as follows: The reliability tests showed that point PTFs produce more accurate estimations than parametric PTFs (see page 10 lines 351-352).**

*Page 9, lines Tables and figures - Tables and figures should be self-explanatory in their titles and contents. The authors should provide all the necessary information such as explanation for abbreviations, measurement units, etc. Please see also specific comments.*
*- The order of captions of tables and figures should generally correspond with the order*
*of appearance in the text:* **All these comments will be addressed in the revised manuscript**

**3. Technical comments**

*Page 1, lines 20-21: the way values of RMSE are reported in the abstract is confusing. Page 1, line 21: RMSE values are in cm3 cm-3 as units for water retention. Page 1, line 28: medium textural class Page 2, lines 44-45: 'estimated' instead of 'constructed' Page 2, line 47: the parameters of the van Genuchten model ($\theta_s$, $\theta_r$, $\alpha$ and n) have been introduced before the van Genuchten model itself (line 51). It would be more logical to remove them in line 47 and insert them in line 53. Page 2, line 53: I would suggest: ': to predict the van Genuchten parameters ($\theta_s$, $\theta_r$, $\alpha$ and n) with soil texture classes only: : :' Page 2, line 54: 'bulk density' instead of 'density' Page 2, line 55: 'Pedotransfer functions' instead of 'PTFs' Page 2, line 60: 'water retention' instead of 'the water retention' Page 2, line 72: 'pedotransfer' should be deleted Page 2, line 76: 'complementary' instead of 'complimentary' Page 3, lines 82-83: I would suggest: 'Deriving and validating two approaches of PTFs using regression methods:' Page 3, line 87: 'input perturbation' is not appropriate. Please rephrase. Page 3, line 92: I would suggest: 'The PTFs are developed using a database of soil samples collected from some regions in Algeria' Page 3, line 93: 'contains' instead of 'containing' Page 3, line 99: 'more than 30 cm' instead of 'upper than 30 cm' Page 3, line 100: 'was conducted using: : :' Page 3, line 100: I would suggest: 'Undisturbed soil samples taken: : :' Page 3, line 101: '(According to the case)' should be deleted Page 3, line 104: I would sug- gest: 'Water content measurements were conducted by the gravimetric method' Page 3, line 106: The word 'defended' is inappropriate. Please rephrase Page 3, lines 111- 112: 'were calculated' instead of 'will be calculated' Page 3, line 117: 'by comparing the values that they predicted' Page 3, lines 118-119: I would suggest: 'To discuss the validity of the PTF developed, we used the following criteria:' Page 3, lines 119-120: 'the root mean square error (RMSE)' instead of 'the mean square error (RMSE)' Page 3, line 120: 'of the quality of the prediction' instead of 'of quality prediction' Page 4, line 129: 'the root mean square error (RMSE)' instead of 'the mean square error (RMSE)' Page 4, lines 143-144: Is it X1 or $X_1$? This needs to be uniform. Page 4, line 144: I would suggest: ': : :X=(X1,: : :, Xp) is the input variable set' Page 4, lines 147-149: The elements of equations 6 and 7 are not explained Page 4, line 157: 'between 0 and 1' instead of 'between [0.1]' Page 5, line 161: What is $X_i^*$ in equation 9? Page 5, line 168: I am missing sentences that introduce Table 1 and Table 2. Moreover, Table 2 is not mentioned in the whole text. Page 5, line 170: I would suggest as title: 'Development of PTFs' Page 5, lines 172-173: With respect to the title in line 170, this sentence should be placed at the end of the paragraph Page 5, line 177: 'soil water retention' instead of 'the soil water retentions' Page 5, line 184: I would suggest: 'the point MLR PTFs' instead of 'the PTF points (MLR)' Page 5, line 185: '0.041 and 0.044' instead of '0,041 and 0,044' Page 5, line 188: 'parametric PTFs' instead of 'parametric PTF' Page 5, line 189: 'neural' instead of 'neuron' Page 5, line 189: 0.0613 and 0.0605 cm3 cm-3 Page 6, line 199: 'while' should be deleted Page 6, line 207: 'PTFs' instead of 'pedotransfer functions' Page 6, line 209: 'fundamental to understand the: : :' instead of 'fundamental to understanding the: : :' Page 6, line 211: 'as bulk density' instead of 'as the bulk density' Page 6, lines 214-217: This sentence is too long and should be divided into 2 sentences.*

*Page 6, lines 215-216: 'at two pressure points' instead of 'in two pressure points' Page 6, line 219: 'in third place' instead of 'in third order' Page 6, line 222: 'MLR' instead of 'linear multiple regression' Page 6, line 224: I would suggest: 'point PTF using MLR is mainly based on: : :' Page 6, line 225: I would suggest: 'parametric PTF using MNLR' Page 6, line 226: I would suggest: 'which has other inputs than tex- ture and bulk density' Page 6, line 229: I would suggest: 'textural grouping' instead of 'textural classification' (see Page 6, line 234) Page 6, lines 231-232: I would suggest: ': : :used to develop PTFs from basic soil characteristics to estimate water retention for different textural classes' Page 6, line 235: 'FAO' instead of 'FOA' Page 6, line 238: I would suggest: 'textural grouping' instead of 'textural stratification' Page 6, line 239: I would suggest: 'a better prediction at -1500 kPa was provided by point PTF' Page 7, lines 242-243: I would suggest: 'explained by difficulties in linking water retention properties of the soil samples with their particle size distribution as...' Page 7, line 244: I would suggest: 'After textural grouping, MLR and MNLR PTFs developed are: : :' Page 7, line 247: I would suggest: 'In the MNLR PTFs' instead of 'Into the MNLR' Page 7, line 259: 'fractions' instead of 'fraction' Page 7, line 260: 'observed' instead of 'observe' Page 7, line 268: 'in all texture classes' instead of 'on all texture classes' Page 7, line 269: 'in the validation dataset' instead of 'in the dataset of validation' Page 7, line 275: 'increases' instead of 'increase' Page 7, line 280: 'low sand content' instead of 'small sand content' Page 8, line 283: 'This is the second most influential variable' Page 8, line 285: 'bulk density' instead of 'the bulk density' Page 8, line 289: 'highly related' instead of 'hugely related' Page 8, line 291: Vertisols Page 8, lines 299-300: 'has a major influence' instead of 'is a major influence' Page 8, lines 301-302: I would suggest: 'predicted values very close to the experimental results are obtained' Page 8, line 303: I would suggest: 'depends on the type of regression techniques' Page 8, line 309: 'PTFs' instead of 'PTF' Page 8, line 312: 'with Clay (%)' instead of 'with the Clay (%)' Page 8, line 315: 'than' instead of 't' Page 8, line 319: I would suggest: 'at high and medium soil water potentials' Page 8, lines 321-322: I would suggest: 'textural grouping' Page 9, line 323: I would suggest: 'The lowest values were recorded' Page 9, line 324: 'of' should be deleted Page 9, lines 330-331: I would suggest: 'textural grouping' Page 9, lines 331-332: I would suggest: 'by the poor OM content in the Algeria soil samples' Page 9, lines 331-332, 333-334: Sometimes 'OM', sometimes 'organic matter'. Please be consistent Page 9, line 333: I would suggest: '...water retention. Danalatos et al. (1994) attributed it to...' Page 9, lines 337-338: I would suggest: 'to predict _s values' instead of 'to predict saturated soil water contents' Page 9, line 348: 'Indeed' should be deleted Page 9, line 354: I would suggest: 'textural grouping' Page 9, line 355: 'classes' instead of 'class' Page 10, lines 359-360: I would suggest: with clay content > 60% Page 10, line 361: I would suggest: 'textural grouping' Tables and figures Tables Table 1: - I would suggest as title: 'Soil characteristics of the development and validation datasets' - I would suggest PSD (particle size distribution) instead of Granulometry - 'CV: coefficient of variation' is reported three times on the same table Table 2: - the line separating MLR and MNLR PTFs is not at the right place - 'Point PTFs' instead of 'Points PTF' - 'multiple R2' instead of 'R2 multiple' - a, b, c,...j are not clearly explained. It would be good to write a general equation with a, b, c,...j as coefficients for more clarity - _ should be in kPa-1 - (respectively) should be deleted Table 3: - I would suggest as title: 'Evaluation criteria of water retention pedotransfer functions at -33 kPa and -1500 kPa' Table 4 is missing Table 5: - I would suggest as title: 'Variation of first order sensitivity index along different textural classes' - What does 'Abs' mean? Table 6: - I would suggest as title: 'Pearson correlation matrix between basic soil characteristics in the validation dataset of 53 soil samples' Figures Each figure caption should be located beneath the respective figure Figures 1, 2, 4, 5 and 6: ',' should be replaced by '.' Figure 2: - I would suggest as title: 'Particle size distribution of xx soil samples from Algeria according to the FAO textural triangle' Figure 3: - Figure 3 is not mentioned in the text Figure 4: - I would suggest as title: 'Root mean square error (RMSE) values calculated for different textural classes' Figure 5: - I would suggest as title: 'Variation of first sensitivity index with RMSE after textural grouping' Figure 6: - Police sizes on the 2 graphs are not the same - 'Point PTF' instead:* **All these comments will be addressed in the revised manuscript.**

**Additional References**

**Aubert, G. : Méthodes d'analyses des sols, Edit.C.R.D.P. Marseille, 189, 1978.**
**Botula, Y. D., Van Ranst, E., Cornelis, W.M.: Pedotransfer functions to predict water retention of soils from the humid tropics: A review, Brazilian Journal of Soil Science, 38, 679-698, 2014.**
**BRUAND, A., PEREZ-FERNANDEZ, P., DUVAL, O., QUETIN, P., NICOULLAUD, B., GAILLARD, H., RAISON, L., PESSAUD, J.F. and PRUD'HOMME, L. : Estimation des propriétés de rétention en eau des sols: utilisation de classe de pédotransfert après stratifications texturale et texturo-structurale. Etud. Gest. Sols, 9:105-125, 2002.**

Mirus, B. B.,: Evaluating the importance of characterizing soil structure and horizons in parameterizing a hydrologic process model, Hydrological Processes, 29(21), 4611-4623, 2015.

Pachepsky. Y. A., Rawls, W.J.: Soil structure and pedotransfer function,Eur,J,Soil Sci, (54), 443-452,2003

Richards, L. A., and Fireman, M..: Pressure-plate apparatus for measuring moisture sorption and transmission by soils. Soil Sci. 56: 395–404, 1943.

Robinson, G. W.: A new method for the mechanical analysis of soils and other dispersions. J. Agric. Sci., 12, 1922.

Stumpp, C., Engelhardt, S., Hofmann, M., and Huwe, B.: Evaluation of Pedotransfer Functions for Estimating Soil Hydraulic Properties of Prevalent Soils in a Catchment of the Bavarian Alps, European Journal of Forest Research, 128(6), 609-620, 2009.

Vereecken, H., Feyen, J., Maes, J. and Darius, P.: Estimating the soil moisture retention characteristic from texture, bulk density, and carbon content, Soil Sci, 148,389-403, 1989.

Walkley, A. and Black, I.A.: An examination of the Degtjareff method for determining organic carbon in soils: Effect of variations in digestion conditions and of inorganic soil constituent, Soil Sci, 63, 251-263, 1934.

Winfield, K.A., Nimmo, J.R., Izbicki, J.A., and MartinResolving, P.M.: Structural Influences on Water-Retention Properties of Alluvial Deposits, Vadose Zone Journal, (5), 706-719, 2006.

**Dear Dr David Dunkerley (Topical Editor),**

**Following is the revised version of manuscript. Changes have been highlighted using Tow different colors based on reviewer comments. Changes based on first reviewer comments are highlighted yellow; changes based on second reviewer comments are highlighted green.**

**Sincerely,**

**Sami Touil and co-authors**

[revised manuscript text omitted]

629 **Figure 4.** Root mean square error (RMSE) values calculated for the different textural classes.

630

631

632

633 **Figure 5.** Variation in first sensitivity index with RMSE after textural grouping.

634

[Figure]

635

636 **Figure 6.** Scatter plots of measured soil water retention versus predicted soil water retention.

637

---

## Editor Decision (ED1)

**Reviewer #1:**

SOIL Discuss doi:10.5194/soil-2016-18, 2016 Sensitivity analysis of point and parametric pedotransfer functions for estimating soil water retention

Overall opinion about the revised manuscript

The authors made a great effort to improve the quality of the manuscript. Therefore, the article can be accepted for publication with minor revisions since there is still a limited number of issues that need to be addressed by the authors before publication. These are reported in the specific comments:

Specific comments:

Page 2, line 55: I would suggest: 'Some 97% of water retention PTFs for soils in the tropics'

Page 3, line 92: In the manuscript, the authors start by Figure 3a instead of Figure 1.

Page 4, line 124: An isolated 'T' should be removed.

Page 4, line 146: I would suggest: 'as locally developed PTFs do' instead of '...as well as the locally developed PTFs'.

Page 5, line 161: After some inquiry, it seems that the original formula of the index of Willmott has no '1/n' at the numerator. The authors should check this. Page 6, line 213: I would suggest 'ME values close to zero'

Page 6, line 224: There is an issue with the order of citing Figures in the text. The authors are jumping from Figure 3b (page 3, line 108) to Figure 6.

Page 6, line 224: The index of agreement results are shown in Table 3 and not in Figure 6.

Page 6, lines 228-229: This paragraph should not be isolated.

Page 7, line 258: Figure 4 illustrates the RMSE values (see the next sentence) and not the improvement of estimation of PTFs after textural grouping.

Page 8, line 279: RMSE 0.030 cm3 cm-3

Page 8, line 280: ']' should be replaced by ')'

Page 8, line 295: I would suggest: '...increasing accuracy in PTFs at -33 kPa after fixing the clay content.'

Page 8, line 301: '(%)' should be removed.

Page 8, line 304: I am missing something here. What 'they' refer to? Anyhow I would expect that soils in the medium textural class would drain water more quickly than in the very fine and fine classes.

Page 9, lines 316-318: This long sentence should be divided into two separate sentences to make sense.

Page 9, line 319-320: I would suggest: 'The accuracy of quality estimation at - 33 kPa in the medium class when fixing the BD for the two PTF approaches...'

Page 9, lines 322-324: I asked the authors to reformulate this sentence and I still do not agree with the rephrasing with regards to Vertisols. Is it really what

Rawls et al. (2003) stated? Vertisols are well-known to be swelling-shrinking soils with high clay content and high water content in wet conditions.

Page 9, line 328: I would remove 'capillary'

Page 9, line 333: The structural information Nguyen et al. (2015) are talking about is actually related to the more categorical (i.e. qualitative) soil structure information and not to bulk density.

Page 9, line 335: 'is related' instead of 'related'

Page 9, line 349: 'soils' instead of 'soil'

Page 10, lines 359-360: of quality estimation 'at -33 kPa' in the medium class?

Page 10, line 360: 'at -33 kPa' should be removed at the end of the corrected sentence (see previous comment).

Page 10, line 375: 'Revista Brasileira de Ciencia do Solo' instead of 'Brazilian Journal of Soil Science'.

Page 10, lines 378-379: The names of the authors should not be in full capital letters.

Page 17, line 586: Commas should be replaced by points in Figures 1, 2, 4, 5 and 6.

Page 17, lines 592-596: Figure 3. Ideally, the two textural triangles should be of the same shape to allow a direct comparison.

**Reviewer #2:**

p.3: BD, OM, PSD should be defined at their first us

p.5: Equation (6) is still \*wrong\*: it does not have the term 1/n. Does this modification affect the results?

p.5: better link equation (7) and (8): they are just presented without any note line 335: most insignificant -> less significant

Moreover I recommend again to \*present somewhere\* the obtained PTF, e.g. in general form with a table of the coefficient. p.3: BD, OM, PSD should be defined at their first us

p.5: Equation (6) is still \*wrong\*: it does not have the term 1/n. Does this modification affect the results?

p.5: better link equation (7) and (8): they are just presented without any note line 335: most insignificant -> less significant

Moreover I recommend again to \*present somewhere\* the obtained PTF, e.g. in general form with a table of the coefficient.

---

## Author Response (AR2)

Dear Dr David Dunkerley (Topical Editor),

Thank you for the comprehensive and thorough review. We really appreciate the time and effort of the Topical Editor and reviewers put in on their feedback. We have considered all their suggestions and we feel it has greatly improved the manuscript.

Please find below our response to the reviewers' comments. Reviewer' comments are in italic, the authors' replies are in bold.

*Reviewer #1:*

*SOIL Discuss doi: 10.5194/soil-2016-18, 2016*
*Sensitivity analysis of point and parametric pedotransfer functions for estimating soil water retention*
*Overall opinion about the revised manuscript*
*The authors made a great effort to improve the quality of the manuscript. Therefore, the article can be accepted for publication with minor revisions since there is still a limited number of issues that need to be addressed by the authors before publication. These are reported in the specific comments:*

*Specific comments:*

*Page 2, line 55: I would suggest: 'Some 97% of water retention PTFs for soils in the tropics'.* **It will be done as suggested by the reviewer (see page 2, lines 55)**
*Page 3, line 92: In the manuscript, the authors start by Figure 3a instead of Figure 1.* **The reviewer is right. The order of all figures will be addressed in the revised manuscript.**
*Page 4, line 124: An isolated 'T' should be removed. Indeed.* **It will be removed.**
*Page 4, line 146: I would suggest: 'as locally developed PTFs do' instead of '…as well as the locally developed PTFs'.* **It will be done as suggested by the reviewer (see page 4, lines 149-150)**.
*Page 5, line 161: After some inquiry, it seems that the original formula of the index of Willmott has no '1/n' at the numerator. The authors should check this.* **Indeed. It was a typing error. It will be corrected (see page 4, lines 149-150).**
*Page 6, line 213: I would suggest 'ME values close to zero'.* **It will be modified (see page 6, lines 215).**
*Page 6, line 224: There is an issue with the order of citing Figures in the text. The authors are jumping from Figure 3b (page 3, line 108) to Figure 6.* **The reviewer is right. The order of all figures will be addressed in the revised manuscript.**
*Page 6, line 224: The index of agreement results are shown in Table 3 and not in Figure 6.* **It will modified (see page 6, lines 228).**
*Page 6, lines 228-229: This paragraph should not be isolated.* **It will be addressed (see page 6, lines 231-232).**
*Page 7, line 258: Figure 4 illustrates the RMSE values (see the next sentence) and not the improvement of estimation of PTFs after textural grouping.* **It will be modified as follows: The results showed that after the textural grouping, there was an improvement in the quality estimation of PTFs only in the medium class. A better prediction at -1500 kPa was provided by point PTFs (RMSE = 0.027 cm3 cm-3) and parametric PTFs (RMSE = 0.038 cm3 cm-3) at -1500 kPa (Figure 5). (See page 7, lines 261-265).**
*Page 8, line 279: RMSE 0.030 cm3 cm-3.* **It will be addressed (see page 8, lines 284).**
*Page 8, line 280: ']' should be replaced by ')'.* **It will be addressed (see page 8, lines 286).**
*Page 8, line 295: I would suggest: '...increasing accuracy in PTFs at -33 kPa after fixing the clay content.'.* **It will be done as suggested by the reviewer (see page 8, lines 300-301)**
*Page 8, line 301: '(%)' should be removed.* **It will be removed (see page 8, lines 307).**
*Page 8, line 304: I am missing something here. What 'they' refer to? Anyhow I would expect that soils in the medium textural class would drain water more quickly than in the very fine and fine classes.* **It will be removed.**
*Page 9, lines 316-318: This long sentence should be divided into two separate sentences to make sense.* **It will be modified as follows: this is the second most influential variable on the point PTF (MLR) response on all textural class. The important variation of sensitivity index is noted mainly in the very fine textural class at -33 kPa ($V_{Si}$ = - 50, 5%).**

*Page 9, line 319-320: I would suggest: 'The accuracy of quality estimation at - 33 kPa in the medium class when fixing the BD for the two PTF approaches.... '.* **It will be done as suggested by the reviewer (see page 9, lines 327-328).**

*Page 9, lines 322-324: I asked the authors to reformulate this sentence and I still do not agree with the rephrasing with regards to Vertisols. Is it really what Rawls et al. (2003) stated? Vertisols are well-known to be swelling-shrinking soils with high clay content and high water content in wet conditions. Rawls et al. (2003) stated? Vertisols are well-known to be swelling-shrinking soils with high clay content and high water content in wet conditions.* **It will be removed.**

*Page 9, line 328: I would remove 'capillary'.* **It will be removed.**

*Page 9, line 333: The structural information Nguyen et al. (2015) are talking about is actually related to the more categorical (i.e. qualitative) soil structure information and not to bulk density.* **It will be removed.**

*Page 9, line 335: 'is related' instead of 'related'.* **It will be addressed.**

*Page 9, line 349: 'soils' instead of 'soil'.* **It will be addressed.**

*Page 10, lines 359-360: of quality estimation 'at -33 kPa' in the medium class?*

*Page 10, line 360: 'at -33 kPa' should be removed at the end of the corrected sentence (see previous comment).* **It will be corrected (see page 10, line 368).**

*Page 10, line 375: 'Revista Brasileira de Ciencia do Solo' instead of 'Brazilian Journal of Soil Science'.* **It will be modified (see page 10, line 384-385).**

*Page 10, lines 378-379: The names of the authors should not be in full capital letters.* **It will be addressed.**

*Page 17, line 586: Commas should be replaced by points in Figures 1, 2, 4, 5 and 6.* **It will be addressed.**

*Page 17, lines 592-596: Figure 3. Ideally, the two textural triangles should be of the same shape to allow a direct comparison.* **It will be modified.**

***Reviewer #2:***

*p.3: BD, OM, PSD should be defined at their first us. It will.* **It will be modified (see page 3, lines 93 and line 103).**

*p.5: Equation (6) is still \*wrong\*: it does not have the term 1/n. Does this modification affect the results?* **Indeed. It was a typing error. It will be corrected (see page 4, lines 149-150).**

*p.5: better link equation (7) and (8): they are just presented without any note.* **It will be addressed.**

*line 335: most insignificant -> less significant.* **It will be modified (see page 9, lines 344).**

*Moreover I recommend again to \*present somewhere\* the obtained PTF, e.g. in general form with a table of the coefficient.* **It will be modified in the revised version. The equations of the obtained PTFs will be presented on table 2 (see page 15, line 546).**

Dear Dr David Dunkerley (Topical Editor),

Following is the revised version of manuscript. Changes have been highlighted using Tow different colors based on reviewer comments. Changes based on first reviewer comments are highlighted yellow; changes based on second reviewer comments are highlighted green.

Sincerely,

Sami Touil and co-authors

[revised manuscript text omitted]

*(\*) S: sand, C: clay, Si: silt, BD: bulk density, OM: organic matter, MLR: Multiple Linear Regression, MNLR: Multiple Non-linear Regression.*

**Table 3.** Evaluation criteria of water retention pedotransfer functions (PTFs) at -33 kPa and -1500 kPa.

| | | | *-33 kPa* | *-1500 kPa* |
|---|---|---|---|---|
| $ME$ $(cm^3\ cm^{-3})$ | *Point PTF* | MLR | 0.0188 | 0.0261 |
| | *Parametric PTF* | MNLR | -0,0016 | -0.0020 |
| | *Rosetta* | H1 | - 0.0902 | -0.0458 |
| | | H2 | - 0.0728 | -0.0436 |
| | | H3 | -0.0991 | -0.0552 |
| $RMSE$ $(cm^3\ cm^{-3})$ | *Point PTF* | MLR | 0.0414 | 0.0444 |
| | *Parametric PTF* | MNLR | 0.0613 | 0.0605 |
| | *Rosetta* | H1 | 0.1170 | 0.0738 |
| | | H2 | 0.0970 | 0.0636 |
| | | H3 | 0.1280 | 0.0749 |
| $d$ $(cm^3\ cm^{-3})$ | *Point PTF* | MLR | 0.9975 | 0.9911 |
| | *Parametric PTF* | MNLR | 0.9938 | 0.9775 |
| | *Rosetta* | H1 | 0.9623 | 0.9427 |
| | | H2 | 0.9775 | 0.9597 |
| | | H3 | 0.9519 | 0.9331 |

**Table 4.** Variation of first order sensitivity index ($S_i$) in the different textural classes.

| | | | Si (%) | | S (%) | | C (%) | | BD (g/cm³) | | OM (%) | |
|---|---|---|---|---|---|---|---|---|---|---|---|---|
| | | Tex-class | $V_{Si}$ | A.E | $V_{Si}$ | A.E | $V_{Si}$ | A.E | $V_{Si}$ | A.E | $V_{Si}$ | A.E |
| RML | at -33 kPa | VF | *Abs* | | -1.2 | | -0.4 | | -50.5 | - | 4.6 | |
| | | F | *Abs* | | -43.2 | - | -10.7 | - | -39.9 | - | 0.2 | |
| | | M | *Abs* | | -103.3 | - | -27.5 | + | -44.4 | + | -5.7 | |
| | at -1500 kPa | VF | *Abs* | | -0.3 | | 0.9 | | -27.3 | - | 1.1 | |
| | | F | *Abs* | | -46.2 | - | -20.7 | - | -41.6 | - | 0.1 | |
| | | M | *Abs* | | -86.4 | - | -52.9 | - | -22.9 | - | -2.3 | |
| MNLR | at -33 kPa | VF | 0.4 | | -0.2 | | 0.1 | | -00.1 | | -0.05 | |
| | | F | -1.6 | | -40.9 | - | -1.1 | | -2.5 | | -0.1 | |
| | | M | 15.0 | | -5.2 | | 15.1 | + | 21.6 | + | 22.3 | + |
| | at -1500 kPa | VF | - 4.6 | | -0.3 | | -1.8 | | -1.4 | | -00.5 | |
| | | F | 28.6 | + | 18.9 | - | 4.6 | | 0.4 | | 0.1 | |
| | | M | -36.7 | - | -16.7 | - | -22.6 | - | 8.9 | | -8.4 | |

*Abs: absent in the model, V Si: variation first sensitivity index; A.E.: improving estimation.*

**Table 5.** Pearson correlation matrix between basic soil characteristics in the validation dataset of 53 soil samples.

| Variables | S$_i$ (%) | C (%) | S (%) | BD (g/cm$^3$) | OM (%) |
|---|---|---|---|---|---|
| Si% | **1** | | | | |
| S % | **-0.334** | **1** | | | |
| C % | -0.159 | **-0.878** | **1** | | |
| BD (g/cm3) | 0.164 | -0.185 | 0.11 | **1** | |
| OM (g/100g) | -0.174 | -0.166 | 0.263 | -0.19 | **1** |

*The values in bold differ from 0 to a level of significance α = 0.05.*
*Si: silt, S: sand, C: clay, BD: bulk density, OM: organic matter*

**Figures:**

[Figure]

**Figure 3̶1̶—̶.̲ ̲**(a): Texture fractions of dataset (242 samples) based on USDA system. (b): Particle size distribution of 53 soil samples from Algeria according to FAO ̲-textural triangle ̲(FAO, 1990)̲.

[Figure]

**Figure 2̶.̲** Scatter plots of measured versus predicted soil water retention by Rosetta H2.

[Figure]

**Figure 3..** Scatter plots of measured soil water retention versus predicted soil water retention.

[Figure]

**Figure 4..** First order sensitivity index.

[Figure]

|  | Very Fine | Fine | Medium |
|---|---|---|---|
| ■RMNL | 0,084 | 0,072 | 0,056 |
| ▤RLM | 0,053 | 0,056 | 0,03 |

|  | Very Fine | Fine | Medium |
|---|---|---|---|
| ■RMNL | 0,089 | 0,06 | 0,038 |
| ▤RLM | 0,06 | 0,048 | 0,027 |

**Figure 54.. .** Root mean square error (RMSE) values calculated for the different textural classes.

[Figure]

**Figure  Variation** in first sensitivity index with RMSE after textural grouping.